# Astrocytic modulation of excitatory synaptic signaling in a mouse model of Rett syndrome

**Benjamin Rakela, Paul Brehm, Gail Mandel\***

Vollum Institute, Oregon Health & Science University, Portland, United States

**Abstract** Studies linking mutations in Methyl CpG Binding Protein 2 (MeCP2) to physiological defects in the neurological disease, Rett syndrome, have focused largely upon neuronal dysfunction despite MeCP2 ubiquitous expression. Here we explore roles for astrocytes in neuronal network function using cortical slice recordings. We find that astrocyte stimulation in wild-type mice increases excitatory synaptic activity that is absent in male mice lacking MeCP2 globally. To determine the cellular basis of the defect, we exploit a female mouse model for Rett syndrome that expresses wild-type MeCP2-GFP in a mosaic distribution throughout the brain, allowing us to test all combinations of wild-type and mutant cells. We find that the defect is dependent upon MeCP2 expression status in the astrocytes and not in the neurons. Our findings highlight a new role for astrocytes in regulation of excitatory synaptic signaling and in the neurological defects associated with Rett syndrome.

DOI: https://doi.org/10.7554/eLife.31629.001

## Introduction

New roles played by astrocytes in nervous system functioning continue to emerge. However, many questions related to the mechanisms underlying astrocyte-neuronal communication, as well as to the general importance of this communication, still remain unanswered. Patch clamp electrophysiology, especially the ability to make simultaneous measurements from the astrocyte and the neuron, provides a sensitive means to probe their interactions. Here, we use dual patch clamp recordings to extend our understanding of astrocyte-mediated modulation of synaptic transmission within the cortical network. This approach also provides a unique opportunity to determine the potential consequences of defective astrocyte-neuronal signaling in a neurological disease, using a mouse model for Rett syndrome (RTT).

Rett syndrome has remained the subject of great interest since the defective gene was first identified as encoding the ubiquitously expressed X - linked transcription factor, Methyl CpG Binding Protein 2, MeCP2 (*Amir et al., 1999*). Affected males have a severe brain disorder and usually die prior to the age of two. Affected females are mosaic due to dosage compensation by X chromosome inactivation and live on average to ~40 years of age. The disease in females is characterized by regression of early developmental milestones, such as speech and purposeful hand motions, followed by the acquisition of multiple motor abnormalities, anxiety and seizures (*Chahrour and Zoghbi, 2007*). Several early studies using RTT mouse models suggested that the symptoms were due exclusively to loss of MeCP2 from neurons. For example, expression of *Mecp2* from the neuronal *tau* locus in mice prevented the appearance of some RTT-like phenotypes (*Luikenhuis et al., 2004*). More recently, loss and restoration of MeCP2 just in inhibitory (*Ure et al., 2016*) or excitatory neurons (*Meng et al., 2016*) was shown to affect RTT phenotypes differently. Finally, early immunohistochemical studies indicated that astrocytes did not express MeCP2 (*Akbarian et al., 2001*; *Shahbazian et al., 2002*; *Kishi and Macklis, 2004*; *Jung et al., 2003*). Thus, it was very

**\*For correspondence:**
mandelg@ohsu.edu

unexpected when it was first reported that astrocytes express MeCP2 (*Ballas et al., 2009*) and that restoration of MeCP2 expression specifically in astrocytes, in otherwise null mice, delayed the progression of RTT- like symptoms (*Lioy et al., 2011*). These, and other findings (*Maezawa et al., 2009*; *Turovsky et al., 2015*; *Yasui et al., 2017*; *Delépine et al., 2016*; *Garg et al., 2015*; *Williams et al., 2014*; *Nguyen et al., 2012*; *Okabe et al., 2012*; *Lioy et al., 2011*) suggested that astrocyte-neuronal communication had an important physiological role to play in vivo, but left open the question of whether MeCP2- deficient astrocytes contributed directly to abnormal neuronal physiology.

Applying dual patch clamp recordings to cortical slices from wild-type mice, we find that astrocyte stimulation increases the frequency of both excitatory glutamatergic and excitatory GABAergic neuronal synaptic currents in a calcium-dependent manner. We further determine that in global MeCP2 null mice, this astrocyte-mediated synaptic modulation is absent and calcium signals in astrocytes are severely blunted. Exploiting the mosaic expression of MeCP2 in female RTT mice, which express GFP from the endogenous *Mecp2* locus in wild-type cells, we find that astrocyte-neuronal signaling proceeds normally only when the astrocyte expresses MeCP2. Our results provide the first cellular support for the idea that normal astrocytes can restore at least one aspect of excitatory synaptic activity to mutant neurons in a RTT brain.

## Results

### Astrocytes mediate increases in cortical neuronal signaling

Dual patch clamp recordings were performed on layer II/III pyramidal neurons and neighboring astrocytes in brain slices derived from the barrel cortex of postnatal (p10-12) wild-type mice (*Figure 1a–b*). An astrocyte was first selected on the basis of red fluorescence following incubation of the slice in 100 nM sulforhodamine 101 (SR101; *Figure 1c*). Then, a superficial proximal neuron was selected on the basis of somal morphology and size using DIC optics (*Figure 1d*). Once the whole cell voltage clamp mode was established, cellular identity was confirmed on the basis of input resistance, which is four fold higher for neurons (143 ± 24 MΩ) compared to astrocytes (35 ± 12 MΩ).

When held at −80 mV, both neurons and astrocytes could be further distinguished by spontaneous synaptic events (*Figure 2*). Neuronal recordings were associated consistently with spontaneous synaptic events that were all inwardly directed (*Figure 2a*). The spontaneous synaptic events ranged in frequency from 99 ± 39 events/m to 537 ± 147 events/m and a corresponding amplitude range of 4.6 ± 2 pA to 363 ± 122 pA for the smallest and largest events in each recording. The smallest amplitude events detected were limited by the noise levels, which differed between recordings. By contrast, all astrocytes were quiescent, including during the time periods directly following the depolarization that was used to activate synaptic responses in the proximal neuron.

Depolarization of the astrocytes using 20 consecutive 10 ms voltage steps from −80 mV to +60 mV led to an increase in the frequency of the synaptic currents in the proximal, paired neuron (*Figure 2a–b*). The increase in event frequency occurred within the first 5 s following the onset of astrocyte depolarization, with the peak increase occurring within 40 s as indicated by the color-coded asterisks (*Figure 2b*). The response decayed from peak during the ensuing 60 s post-stimulation period and returned to pre-stimulus levels within an additional 60 s (*Figure 2b*). Cumulative data from neuronal recordings indicated a significant increase in the frequency of synaptic currents following astrocyte depolarization (pre-stim: 272 ± 180 events/m, post-stim: 337 ± 143 events/m, p=0.002, n = 14) (*Figure 2c*), which corresponded to an average 1.48 ± 0.58 fold increase in frequency (*Figure 2d*). The pooled amplitude distributions for pre- and post- stimulation bore similar distributions (*Figure 2e*) and, when expressed as fold-change for individual experiments, there was no overall difference in the average amplitude before and after stimulation (pre-stim: 31 ± 24 pA, post-stim: 30 ± 27 pA; 1.02 ± 0.28 fold change; *Figure 2f*). To ensure that the absence of amplitude changes was not biased by event number differences between recordings, event number versus amplitude histograms were first generated for each recording and normalized to peak event number bin. Then, then the normalized individual recordings were summed (*Figure 2—figure supplement 1a*). Statistical comparisons of pre- and post-stimulus amplitudes confirmed no significant differences.

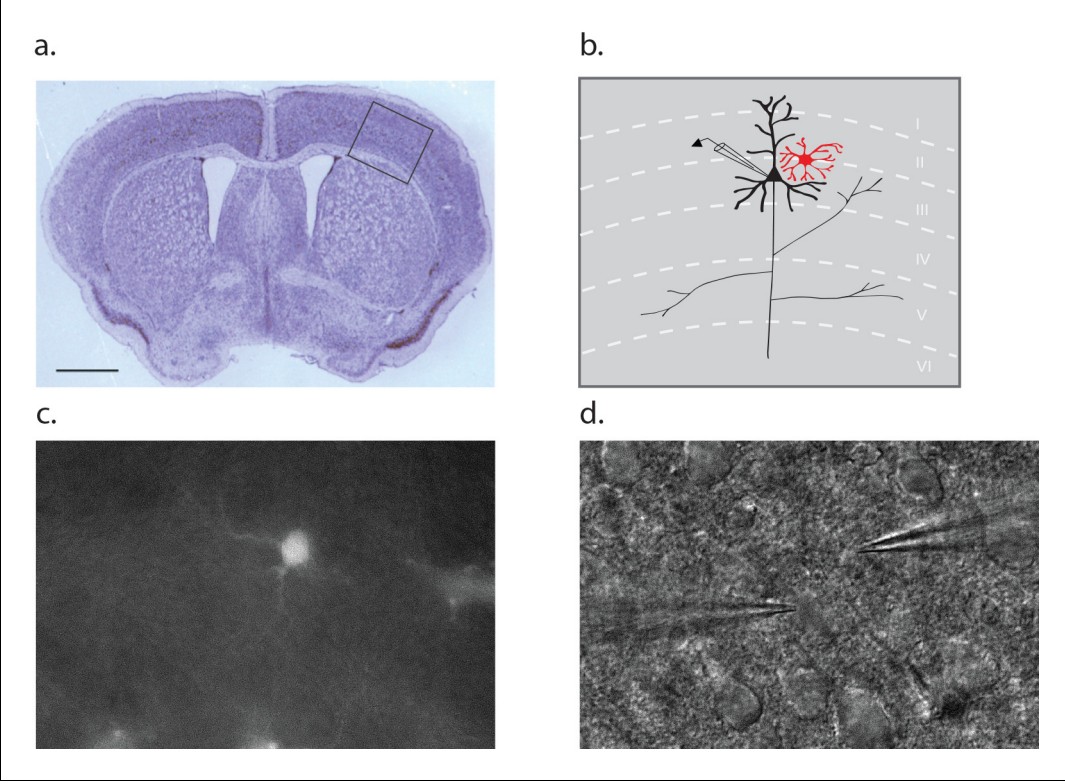

**Figure 1.** Dual patch clamp recording configuration. (**a**) Cross section of a p10 mouse brain through the barrel cortex region with an inset indicating the region from which astrocyte-neuron pairs were obtained for recording. Scale bar 1 mm. (**b**) Schematic representation of the inset area in A, noting the position of a neuron (black) and astrocyte (red) within the layers of the cortex. (**c**) Fluorescence image of a cortical slice showing an SR101 loaded astrocyte. (**d**) The counterpart DIC image showing a typical recording configuration with the neuron electrode on the left and astrocyte electrode on the right.

DOI: https://doi.org/10.7554/eLife.31629.002

To activate the astrocytes in a more physiological manner, we turned to exploiting Protease-activated receptor 1 (Par1). Par1 is expressed in astrocytes at both the mRNA (http://astrocyternaseq.org/addgene?query=F2r), and protein levels (*Junge et al., 2004*). In our hands, immuno-labeling of barrel cortex with a Par1 antibody resulted in fluorescence that was associated exclusively with GFAP-positive astrocytes, with no measurable labeling in wild-type NeuN-positive neurons (*Figure 3a*). Par1 is activated normally by serine proteases and by the synthetic peptide agonist, TFLLR, resulting in Gq-coupled release of intracellular calcium (*Hollenberg et al., 1997*; *Ubl and Reiser, 1997*).

We delivered TFLLR locally to astrocytes by applying low pressure to a 500 µM TFFLR containing patch pipette. As an additional means to limit TFLLR activation to a single astrocyte, we selected SR101 positive astrocytes that were sufficiently separated from one another to be out of range for simultaneous puffer activation. Like depolarization, the application of 500 µM TFFLR ($EC_{50}$ = 1.9 µM) to wild-type astrocytes consistently resulted in an increase in synaptic event frequency in the proximal neuron recording, qualitatively similar to that seen with astrocyte depolarization (*Figure 4a*). Also like depolarization, the increase in synaptic events increased within five seconds of agonist application, and peaked within 35 s (*Figure 4b*). Moreover, a significant increase in synaptic event frequency was recorded for astrocyte-neuron pairs tested (pre-TFLLR: 169 ± 81 events/m, post-TFLLR: 254 ± 116 events/m, p=0.004, n = 10) (*Figure 4c*). When expressed as fold change, an average 1.59 ± 0.16 fold increase was observed (*Figure 4d*). Again, as in the depolarization paradigm there was no change in amplitude pre- and post-TFLLR application (*Figure 4e–f*). Also, normalized distribution plots, as performed for the depolarization data, showed no significant difference between pre- and post TFLLR treatments (*Figure 2—figure supplement 1b*).

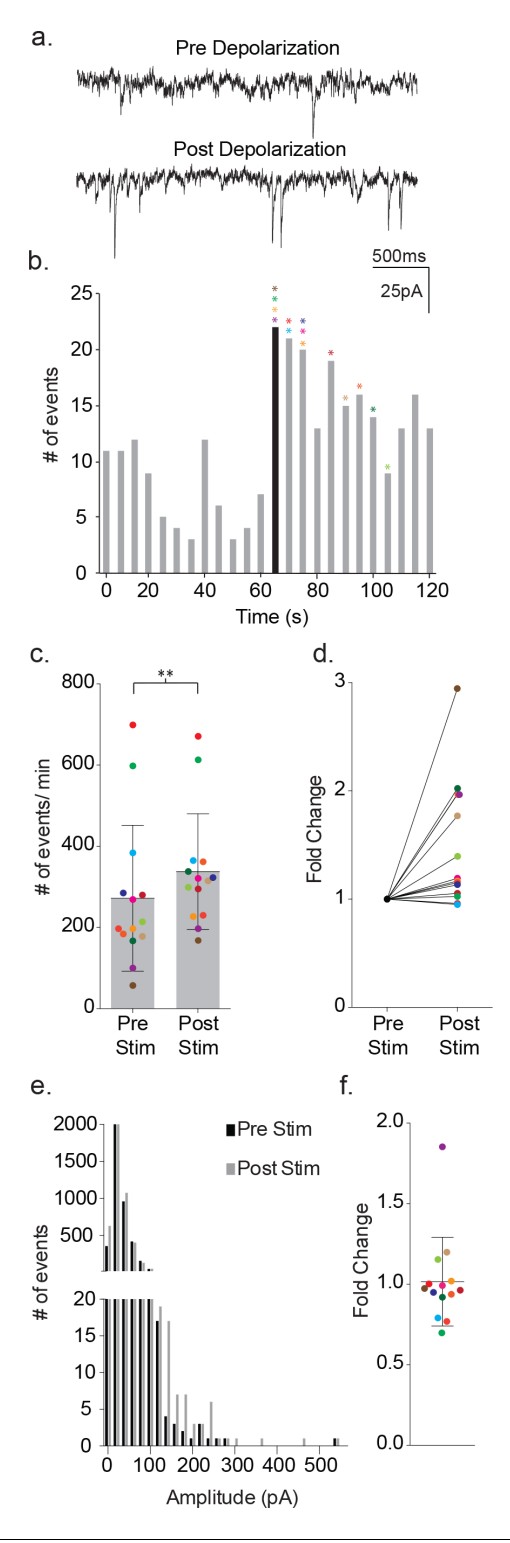

**Figure 2.** Depolarization of astrocytes leads to increased frequency of synaptic currents in cortical neurons of wild-type mice. (a) Representative traces (3 s) of synaptic currents recorded from a wild-type neuron before (top) and after (bottom) administrating 20, 10 ms steps from −80 mV to +60 mV to the

*Figure 2 continued on next page*

# Astrocytes activate excitatory glutamatergic and GABAergic signaling in neurons

The principle neurotransmitters in the cortex are glutamate and GABA, which can be distinguished based upon differences in the ion selectivity of the receptor channel types. In all data shown thus far, the trans-membrane chloride concentration was symmetrical. Therefore, the nonselective GluR and chloride-selective $GABA_A$ receptors share similar reversal potentials (~0 mV), rendering both excitatory. To test whether the $GABA_A$ receptors ($GABA_A$R) are excitatory or inhibitory under native transmembrane chloride gradients, we recorded GABA-activated responses using the perforated patch technique with the chloride impermeant gramicidin as the pore forming antibiotic (*Akaike, 1996*; *Tajima et al., 1996*; *Owens et al., 1996*). Recordings from 16 neurons revealed that application of 50 µM GABA depolarized the neuron to levels sufficient for action potential generation (*Figure 5a*). The depolarizing effect of GABA in these recordings was reversibly inhibited following treatment with 20 µM gabazine (*Figure 5b–c*). When held at 0 mV under current clamp, 50 µM GABA application led to a hyperpolarization (*Figure 5d*), which was reversibly inhibited following treatment with 20 µM gabazine (*Figure 5e–f*), all pointing to an excitatory profile at this developmental stage (p10-12).

The differences in ion selectivity provided a direct way to determine the individual contributions of GABA- and glutamate-mediated currents to the recorded synaptic currents. For this purpose, we dialyzed the neuron with a low chloride containing internal solution to shift the $GABA_A$R reversal potential from ~0 mV to ~−70 mV. We then held the membrane potential of the neuron between the reversal potentials for the two different receptor types (−35 mV), thereby creating inwardly-directed glutamate and outwardly-directed GABA activated currents (*Figure 6a*). Assignment of the inward and outward currents was confirmed with pharmacology. Under basal conditions, 86% of the outward events were inhibited by 20 µM gabazine (pre-gabazine: 18 ± 8 events/m, post-gabazine: 2.6 ± 1.3 events/m; p=0.008, n = 8) with no significant contribution by inward currents (pre-gabazine: 101 ± 46 events/m, post-gabazine: 85 ± 35 events/m; p=0.117, n = 8) (*Figure 6—figure supplement 1a*). Inhibition of GluRs with AP5/NBQX (50 µM/5 µM) reduced the inward current frequency 66% (pre-AP5/NBQX: 104 ± 45 events/m, post-AP5/

*Figure 2 continued*

astrocyte. (**b**) The corresponding temporal sequence for wild-type synaptic current occurrence 60 s prior to and 60 s following astrocyte depolarization. The time of stimulation is indicated by the black bin. The colored asterisks denote the bins where the peak frequency occurred for each respective recording shown in c, d and f. (**c**) The cumulative results from wild-type recordings showing the pre- and post-stimulus event frequencies (mean ± SD, n = 14 recordings, p=0.002, 5 males and 4 females, Wilcoxon matched-pairs signed rank test, two-tailed). (**d**) The fold change in frequency for the individual color-coded recordings. (**e**) Amplitude histograms for pooled recordings measured before (black) and after (gray) astrocyte stimulation. (**f**) The fold change in mean event amplitude ± SD for each color-coded recording is indicated.

DOI: https://doi.org/10.7554/eLife.31629.003

The following figure supplement is available for figure 2:

**Figure supplement 1.** Normalized event number versus amplitude histograms.

DOI: https://doi.org/10.7554/eLife.31629.004

NBQX: 35 ± 17 events/m; p=0.0005, n = 12), with no significant contributions from the outward current class (pre-AP5/NBQX: 38 ± 13 events/m, post-AP5/NBQX events/m: 41 ± 18, p=0.386, n = 12) (*Figure 6—figure supplement 1b*).

Using the directional indicators, we re-tested the contributions of each receptor class to the overall TFFLR-stimulated increase in current frequency. In wild-type recordings, TFFLR application increased the frequency of glutamate-mediated currents by an average 1.67 ± 0.43 fold (pre-TFFLR: 147 ± 63 events/m, post-TFFLR: 236 ± 92 events/m, p=0.002, n = 10) (*Figure 6b–c*). GABA-mediated currents were increased 1.53 ± 0.46 fold in 8 out of 10 recordings (pre-TFFLR: 48 ± 23 events/m, post-TFFLR: 69 ± 34 events/m, p=0.009, n = 10; *Figure 6d–e*). These findings for both basal and TFLLR stimulated GABA-mediated and GluR-mediated synaptic events are in agreement with those obtained using symmetrical chloride (*Figure 6—figure supplement 2*). Treatment with 20 µM gabazine reduced the average basal frequency by 41% (pre: 147 ± 27 events/min post: 87 ± 31 events/min; p=0.001, n = 11) and reduced the TFLLR-induced response by 59% (pre: 260 ± 120 events/min, post: 107 ± 41 events/min; p=0.004, n = 9). Addition of a 50 µM AP5 and 5 µM NBQX cocktail to

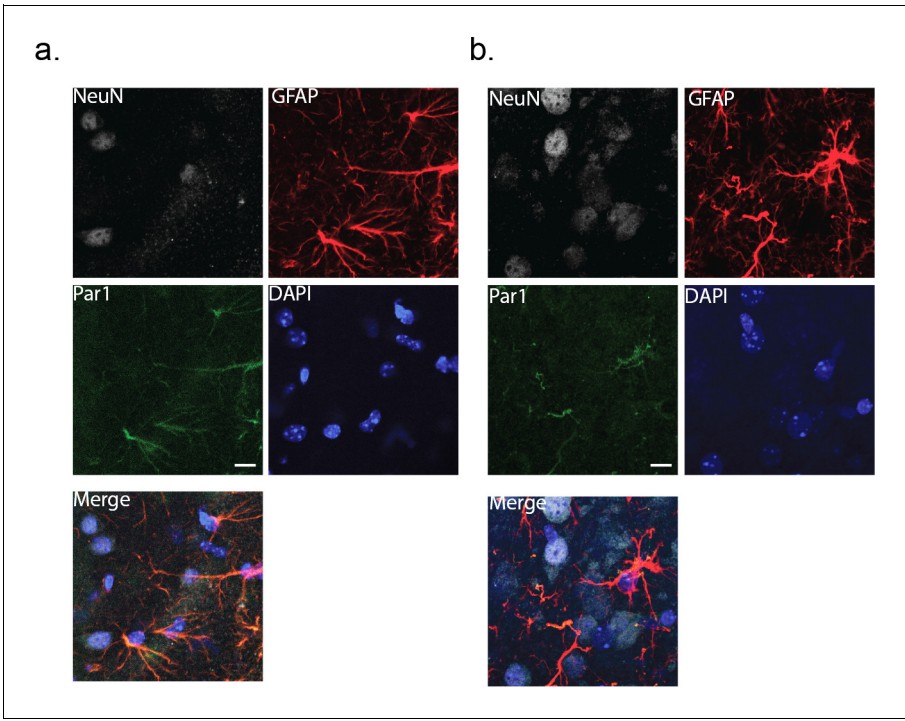

**Figure 3.** Immunohistochemical labeling for the membrane protein, Par1. Cortical slices (80 µm thick) were co-stained with NeuN and GFAP as markers for neurons and astrocytes, respectively. DAPI was used to identify nuclei. (**a**) wild-type cortex (male). (**b**) MeCP2-null cortex (male). Scale bar is 20 µm.

DOI: https://doi.org/10.7554/eLife.31629.005

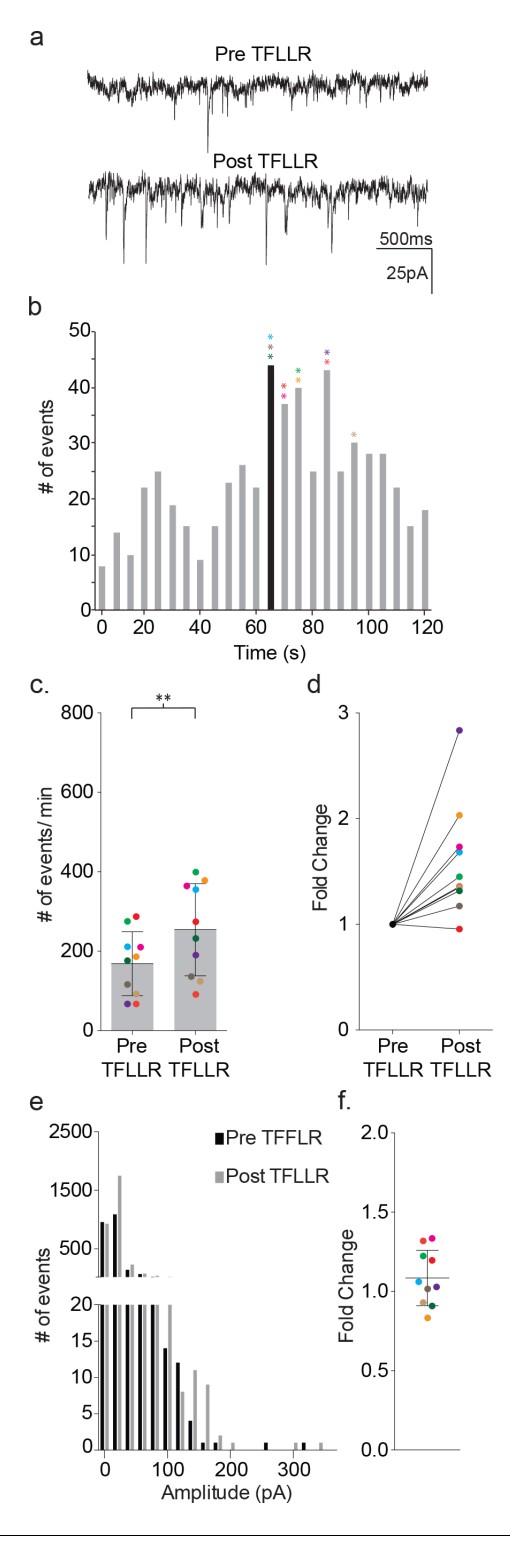

**Figure 4.** TFLLR applied to astrocytes leads to increased frequency of synaptic currents in cortical neurons of wild-type mice. (a) Representative traces (3 s) of synaptic currents recorded from a wild-type neuron before (top) and after (bottom) puffer application of 500 μM TFLLR to the astrocyte. (b) The

block both AMPA and NMDA receptors led to an average 30% reduction in basal event frequency (pre: 121 ± 79 events/min, post: 85 ± 66 events/min; p=0.008, n = 8) and reduced the TFLLR-induced increase in synaptic current frequency by 44% (pre: 120 ± 58/min, post: 67 ± 47/min; p=0.0156, n = 7). Treatment with both inhibitors failed to completely block the inward currents (not shown), also consistent with the incomplete block of inward events observed using asymmetrical chloride.

## Testing for roles of GABA and glutamate release from astrocytes in neuronal activation

Exploiting the inwardly and outwardly directed synaptic currents, we tested whether pharmacological block of either transmitter receptor would block completely the TFLLR activation of the neuronal circuitry involving both transmitter signaling pathways. Treatment with TFLLR and gabazine blocked the outward currents (pre-TFFLR: 9 ± 2 events/m, post-TFFLR: 9 ± 3 events/m, p=0.732, n = 14) (*Figure 6f*), but left untouched the increase in inward event frequency (pre-TFFLR: 169 ± 58 events/m, post-TFFLR: 195 ± 74 events/m, p=0.004, n = 14; *Figure 6f*). This ruled out a requirement of $GABA_A R$ activation for initiation of glutamatergic transmission. By contrast, the application of TFLLR in the presence of the glutamate receptor blockers, AP5/NBQX, failed to invoke statistically significant increases in either inward currents (pre-TFFLR: 89 ± 46 events/m, post-TFFLR: 96 ± 61 events/m, p=0.61, n = 15) or outward currents (pre-TFFLR: 40 ± 25 events/m, post-TFFLR: 52 ± 39 events/m, p=0.124, n = 15) (*Figure 6g*). This result raises the possibility that astrocyte-mediated modulation of neuronal signaling occurs either through direct release of glutamate or release of a substance that acts directly on a glutamatergic neuron.

As a further test of direct release of glutamate or another ionotropic modulator from astrocytes, we tested for any synaptic currents in the proximal neuron that were synchronized to astrocyte stimulation (*Figure 7*). The experimental protocol required that we use astrocyte depolarization because TFLLR application was too slow to test for synchronous release. The astrocyte was depolarized using a single 20 ms pulse to +60 mV at a rate of 1 Hz in order to prevent full activation of the neuronal circuitry that resulted from a stimulus train. This protocol was repeated 100 times for each recording. Pooled pre-stimulus epochs were also obtained

*Figure 4 continued*

corresponding time line for wild-type synaptic current frequency with the time of application indicated by the black bin. The colored asterisks denote the bins where the peak frequency occurred for each respective recording color-coded in c, d and f. (**c**) The cumulative results from wild-type color-coded recordings showing the pre- and post-application event frequencies (mean ± SD, n = 10 recordings, p=0.004, 3 males and 2 females, Wilcoxon matched-pairs signed rank test, two-tailed). (**d**) The fold change for the individual color-coded recordings. (**e**) Amplitude histograms for pooled recordings measured before (black) and after (gray) TFLLR application. (**f**) The fold change in mean event amplitude ± SD for each color-coded recording.

DOI: https://doi.org/10.7554/eLife.31629.006

using the same protocol without depolarization. To improve the signal detection we used symmetrical transmembrane chloride concentration in order to increase the driving force for inward currents. The analysis was restricted to the first 200 ms post-stimulus, the period during which a direct response from the astrocyte would be expected to occur. The pre-stimulus epochs showed events distributed randomly within the 200 ms epoch whereas the pooled post-stimulus intervals showed a non-random distribution with peaks at 40 ms and 60 ms (*Figure 7a*). To determine whether the responses reflected direct release onto the neuron or alternatively indirect signaling through a neuronal circuit, the experiment was repeated in the presence of TTX (*Figure 7b*). This treatment prevented the post-stimulus increase in synaptic event frequency indicating that the events reflected the partial activation of the neuronal circuitry rather than direct astrocytic release. While not as definitive as our pharmacological negation of GABA, the findings argue against direct release of glutamate from astrocytes as the initiator of neuronal synaptic responses.

## Astrocyte-mediated neuronal signaling is absent in MeCP2-deficient mice

Having established the characteristics of astrocyte-mediated activation of neuronal signaling in wild-type mice, we performed paired patch clamp recordings on cortical slices from male MeCP2 globally null mice. The effects of depolarization were examined first (*Figure 8a–f*). Representative traces illustrate the absence of depolarization-activated changes in synaptic current frequency following astrocyte stimulation (*Figure 8a*). The corresponding time course for event frequency also failed to show a significant increase in astrocyte-mediated synaptic current frequency, distinguishing it from wild-type recordings (*Figure 8b*). The absence of astrocyte-mediated effects was reflected in all MeCP2-deficient pairs tested (pre-stim: 260 ± 140 events/m; post-stim: 267 ± 162 events/m; p=0.57; n = 10; *Figure 8c*) (1.01 ± 0.04 fold; *Figure 8d*). Additionally, in the mutant recordings, there were no effects of depolarization on the overall amplitude of synaptic currents (*Figure 8e,f*).

Next, TFLLR was tested for its ability to activate neuronal signaling in slices from MeCP2 null male mice (*Figure 8g–l*). Representative traces prior to and following TFLLR application indicate no increase in event frequency (*Figure 8g*), even though immuno-labeling confirms astrocyte-specific Par1 expression in MeCP2 null cortex (*Figure 3b*) and Western blot analysis shows equivalent Par1 protein levels in MeCP2 null and

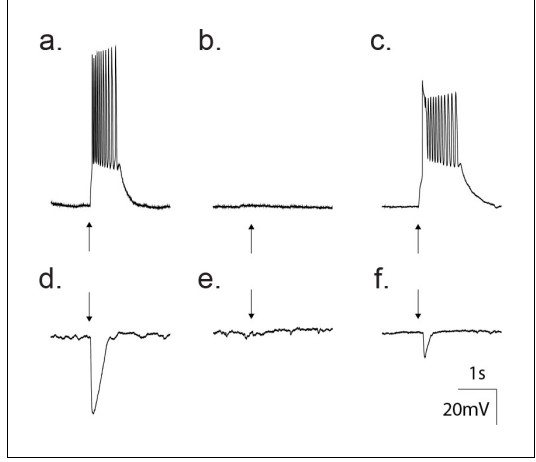

**Figure 5.** Gramicidin perforated patch whole cell recordings from a cortical neuron confirm GABA as an excitatory transmitter at this developmental age. (**a**) Current clamp traces showing that direct application of 50 µM GABA (arrow) depolarized the neuron from −80 mV to threshold for action potentials. (**b**) Treatment with 20 µM gabazine blocked the GABA response. (**c**) Wash out of gabazine restored the ability of GABA to depolarize. (**d**) Current clamp traces where the membrane potential was set to 0 mV. Application of 50 µM GABA (arrow) produced a hyperpolarization. (**e**) The hyperpolarization was blocked by treatment with 20 µM gabazine. (**f**) Wash out of gabazine restored the ability of GABA to hyperpolarize the neuron.

DOI: https://doi.org/10.7554/eLife.31629.007

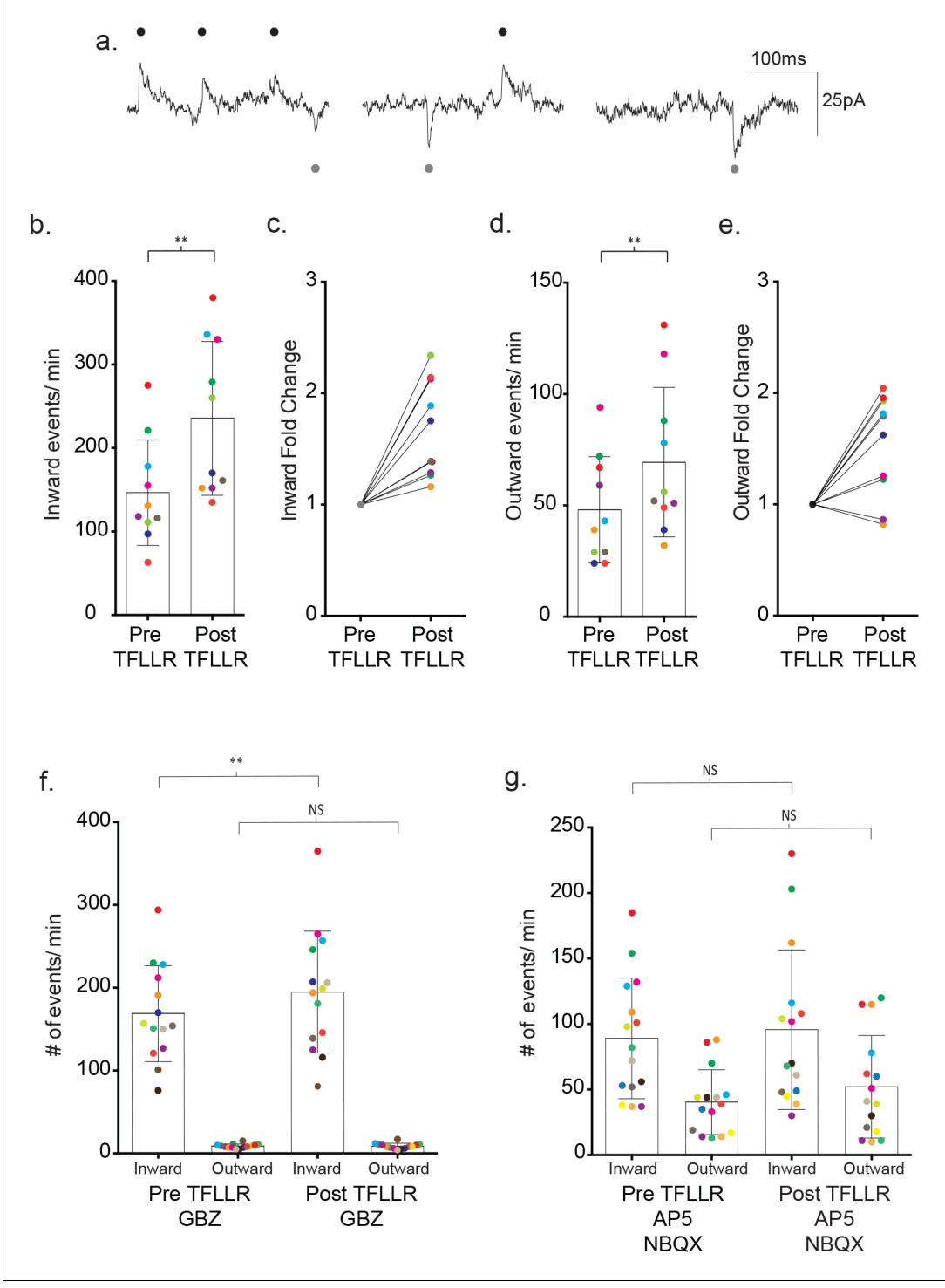

**Figure 6.** Bi-directional currents reveal the individual contributions of GABAergic and glutamatergic signaling. (a) Low chloride internal solution produces inward and outward currents in cortical neurons when held at −35 mV. Three 300 ms representative traces are shown. (b) The frequency of inward events is quantified for each color-coded recording before and after 500 µM TFLLR application, (mean ± SD, p=0.002, n = 10 recordings, 3 males and 2 females, Wilcoxon matched-pairs signed rank test, two-tailed). (c) The corresponding fold change for the individual color-coded recordings. (d) The frequency of outward events for each color-coded recording before and after 500 µM TFLLR application, (mean ± SD. p=0.009, n = 10, 3 males and 2 females, Wilcoxon matched-pairs signed rank test, two-tailed). (e) The fold change for the individual color-coded recordings. (f) The frequency of inward and outward events before and after 500 µM TFLLR application in the presence of 20 µM gabazine (GBZ)

*Figure 6 continued on next page*

*Figure 6 continued*

(inward currents: p=0.004, n = 14 recordings; outward currents: p=0.732, n = 14, 2 males and 2 females Wilcoxon matched-pairs signed rank test, two-tailed). Individual recordings are color-coded. (g) Frequency of inward and outward events before and after 500 µM TFLLR application in the presence of 50 µM AP5 and 5 µM NBQX (inward currents: p=0.61, n = 15 recordings; outward currents: p=0.124, n = 15, 2 males and 3 females Wilcoxon matched-pairs signed rank test, two-tailed). Box and bars indicate the mean ± SD respectively.

DOI: https://doi.org/10.7554/eLife.31629.008

The following figure supplements are available for figure 6:

**Figure supplement 1.** Pharmacological identification of basal, non-stimulated inward and outward currents.
DOI: https://doi.org/10.7554/eLife.31629.009

**Figure supplement 2.** - Effects of GABAR and GluR inhibitors on the frequency of both basal and TFLLR-mediated inward synaptic currents using symmetrical chloride.
DOI: https://doi.org/10.7554/eLife.31629.010

wild-type mice (*Figure 8—figure supplement 1*). As with depolarization, the time course measurements for synaptic events from that recording showed no stimulus-associated increase in frequency (*Figure 8h*). Additionally, the cumulative data for all astrocyte-neuron pairs showed no significant increase in event frequency following agonist application (pre-TFLLR: 262 ± 131 events/m, post-TFLLR: 256 ± 111 events/m, p=0.68, n = 12) (*Figure 8i*). When expressed as fold change, only one of the 12 pairs tested responded positively (*Figure 8j*). Overall, there was no significant increase in event frequency as reflected in the 1.00 ± 0.04 fold change (*Figure 8j*). Finally, no change in amplitude was observed for the pooled events (*Figure 8k*) or in the fold change in amplitude for individual recordings (*Figure 8l*).

To examine more closely for changes in either GABA- or glutamate-mediated event frequency in MeCP2 null recordings, the frequencies were again parsed into inward and outward currents by use of lowered internal chloride. As found with symmetrical chloride, recordings from MeCP2-deficient slices showed no significant increases in either inward or outward current frequency in response to TFFLR (inward: pre-TFLLR: 83 ± 33 events/m, post-TFLLR: 77 ± 32 events/m, p=0.25, n = 6): outward: pre-TFLLR: 25 ± 11 events/m, post-TFLLR: 20 ± 9 events/m, p=0.09, n = 6) (*Figure 8—figure supplement 2*).

## Expression of MeCP2 in astrocytes, not neurons, is required for astrocyte-mediated modulation of neuronal signaling

To determine whether the defective signaling resided in the astrocyte, neuron, or both cell types, we turned to heterozygous mutant female mice that are mosaic for loss of MeCP2. To discriminate between wild-type and mutant cells in the mosaic, we exploited a line of mice that contain a knock-in of GFP into the endogenous *Mecp2* gene. In this line, MeCP2-GFP is expressed in ~50% of all neurons and astrocytes, whereas the remaining 50% of the cells express a truncated, nonfunctional MeCP2 protein (*Lyst et al., 2013*). This line provides a unique opportunity to record from all pair-wise combinations of wild-type (GFP-positive) and MeCP2 null (GFP-negative) astrocytes and neurons.

To ensure that astrocyte-mediated neuronal signaling between wild-type cells in the mosaic line recapitulated the signaling in wild-type mice, we first recorded from pairs where both the astrocyte and neuron expressed MeCP2-GFP (*Figure 9a*). Depolarization of the MeCP2 positive astrocyte increased the frequency of neuronal synaptic events by 1.60 ± 0.23 fold (pre-stim: 150 ± 38 events/m, post-stim: 233 ± 77 events/m, p=0.04, n = 6) (*Figure 9b–c*), recapitulating the findings from wild-type mice. When MeCP2-negative astrocyte/neuronal pairs were tested (*Figure 9d*), the frequency of neuronal synaptic currents did not increase (pre-stim: 147 ± 42 events/m, post-stim: 155 ± 38 events/m, p=0.24, n = 7) (*Figure 9e–f*) similar to the MeCP2 null males. In the third combination (*Figure 9g*), depolarization of a MeCP2-positive astrocyte increased the frequency of synaptic currents in a neighboring MeCP2-negative neuron by 1.66 ± 0.25 fold (pre-stim: 189 ± 02 events/m, post-stim: 268 ± 96 events/m, p=0.02, n = 8) (*Figure 9h–i*). However, when a MeCP2-negative astrocyte was paired with a MeCP2-positive neuron (*Figure 9j*), depolarization failed to increase the frequency of synaptic currents (pre-stim: 176 ± 81 events/m, post-stim: 188 ± 83 events/m, p=0.1176, n = 10) (*Figure 9k–l*). Taken together, these experiments indicate that the astrocyte, not the neuron,

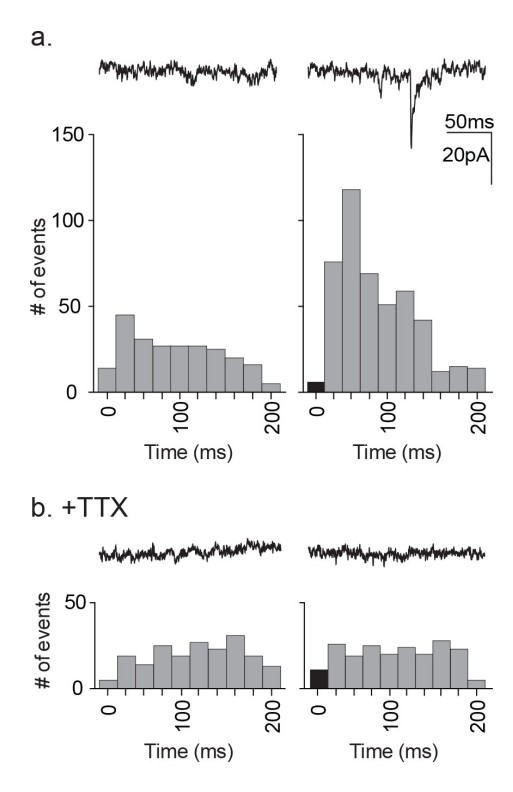

**Figure 7.** Distribution of time-locked neuronal events in response to astrocyte depolarization. (**a**) Two 200 ms representative traces are shown, left: pre-depolarization; right: post-depolarization. Sample frequency histograms from a single recording showing the timing of synaptic events recorded over a 200 ms period prior to (left) and after (right) the astrocyte depolarization (a single 20 ms voltage step to +60 mV). Event number is based on 20 ms bins. Data represents accumulated events from 100 trials, each separated by 1 s (n = 15 neurons, 4 males and 5 females). (**b**) Two 200 ms representative traces are shown, left: pre-depolarization, right: post-depolarization. Sample frequency histograms from a single recording made in the presence of 1 μM TTX (left: pre-depolarization, right: post-depolarization; n = 6 neurons, 2 males and 1 female). Recordings were taken at −100 mV and with symmetrical chloride internal solution.
DOI: https://doi.org/10.7554/eLife.31629.011

requires MeCP2 expression in order to have a significant effect on astrocyte-mediated signaling within the local circuit.

## Intracellular calcium signaling is deficient in MeCP2 null astrocytes

To test whether defects in calcium signaling are involved in the astrocyte-mediated effects shown in this study, we performed several tests: buffering intracellular calcium, imaging intracellular calcium levels and elevating intracellular calcium through uncaging.

First, to determine whether increases in intracellular calcium were required for the astrocyte-mediated signaling with neurons, we dialyzed the astrocyte with a calcium chelator along with fluorescent Alexa Fluor 488. Green fluorescence was monitored to ensure complete dialysis prior to TFLLR application. The intracellular calcium was clamped using either calcium free-10 mM BAPTA (n = 6) or a calcium-BAPTA buffer corresponding to 100 nM free calcium (n = 4). After 15 min of BAPTA dialysis via the astrocyte patch pipette, TFLLR (500 μM) was applied locally while the frequency of events was determined for a proximal neuron. Dialysis with either BAPTA-based buffer reduced the basal frequency of neuronal events from the collective pre-stimulus levels by $0.72 \pm 0.08$ fold, along with no further increase in frequency following TFFLR application (p=0.02, n = 10) (*Figure 10a–b*). This demonstrates a requirement for increased intracellular calcium levels in astrocytes to support the astrocyte/neuronal signaling.

Next, calcium indicators were used to test for defects in intracellular calcium signaling in MeCP2-deficient astrocytes. Wild-type cortical slices were first incubated with Fluo4FF-AM (10 μM; $K_D$ = 9.7 μM) for 35 m, which preferentially loaded SR101 positive astrocytes. In all 23 wild-type astrocytes tested, somatic increases in calcium were observed in response to TFLLR application (*Figure 10c*). The somatic signals were quantitated on the basis of background corrected $\Delta F/F_0$ signals. Wild-type astrocytes produced rises in somatic calcium signals within one second of TFLLR application that required several seconds for complete relaxation to resting values (*Figure 10d–e*). The overall mean for individual experiments corresponded to a $\Delta F/F_0$ of $0.60 \pm 0.56$ (*Figure 10f*; n = 23). Using the global MeCP2 null mice, 23 out of 34 tests of MeCP2 deficient astrocytes produced detectable calcium signals in response to TFLLR application (*Figure 10d–f*). In those 23 cases where responses were detected, the calcium signals followed a time course similar to wild-type (*Figure 10d–e*) but were significantly lower in intensity (mean $\Delta F/F_0$ = $0.27 \pm 0.23$; n = 23) (p=0.012) (*Figure 10d–f*).

Finally, the calcium requirement for astrocyte-mediated modulation of neuronal signaling prompted further tests of whether increasing calcium in a MeCP2 null astrocyte would be sufficient to restore astrocyte-mediated signaling. To do this, we first loaded a wild-type astrocyte with

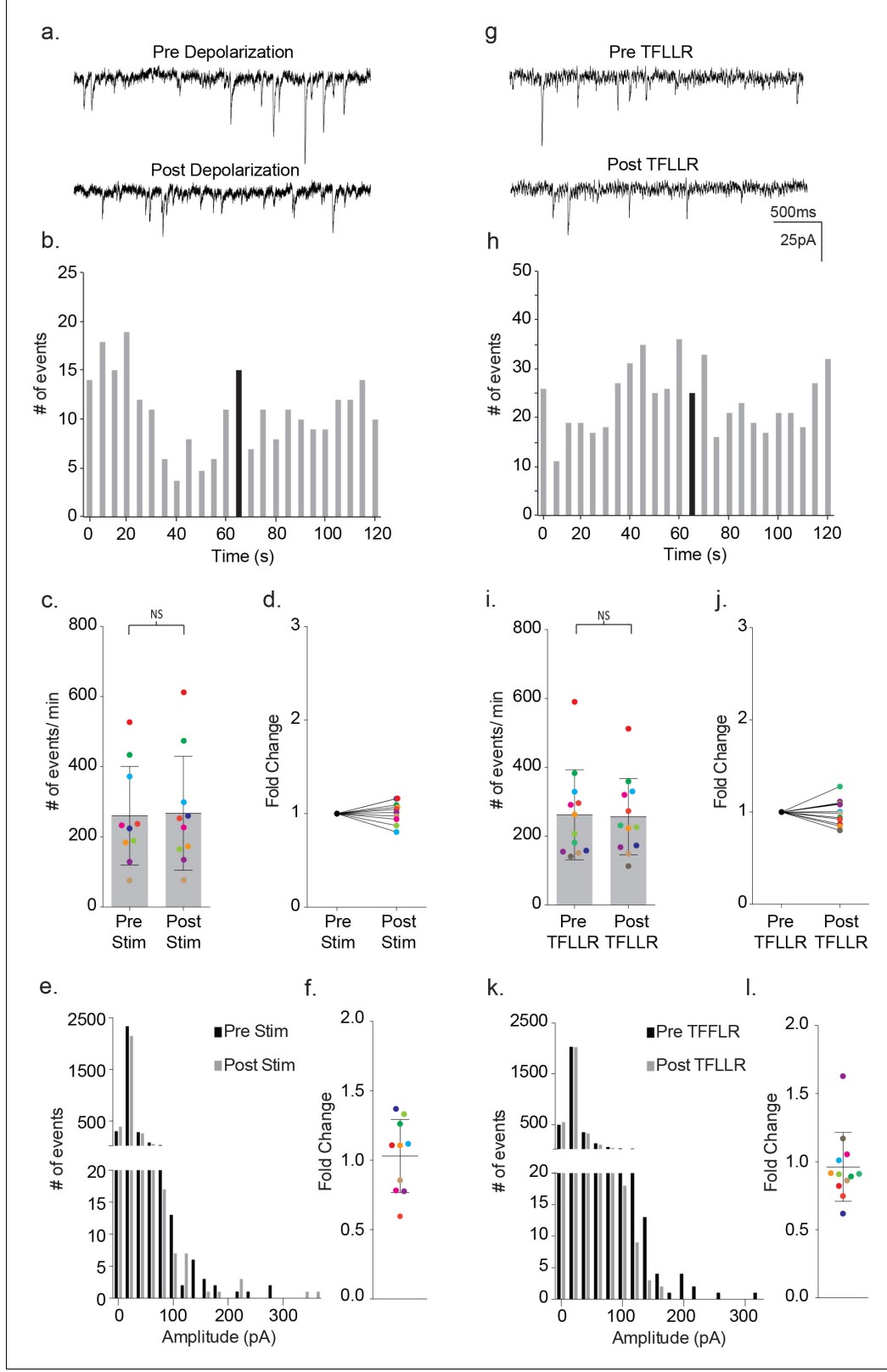

**Figure 8.** Astrocyte-mediated modulation of neuronal signaling is absent in MeCP2-deficient mice. (**a**) Representative traces (3 s) of synaptic currents recorded from a MeCP2 null neuron before (top) and after (bottom) administrating to the MeCP2 null astrocyte 20, 10 ms steps to +60 mV. (**b**) The corresponding temporal sequence for MeCP2 null synaptic currents 60 s prior to and 60 s following astrocyte depolarization. Time of stimulation is indicated by the black bin. (**c**) The results from ten color-coded MeCP2 null recordings (mean ± pre and post-stimulus event frequencies. p=0.57,

*Figure 8 continued on next page*

*Figure 8 continued*

n = 10, 8 males, Wilcoxon matched-pairs signed rank test, two-tailed). (d) The fold change in mean event frequency for the individual color-coded recordings. (e) Amplitude histograms for pooled recordings measured before (black) and after (gray) astrocyte stimulation. (f) The fold change in mean event amplitude for each color-coded recording (mean ± SD). (g, h) Complementary sample recordings, time course histograms obtained from MeCP2 null male mice using TFLLR application. (i) The results from 12 color-coded MeCP2 null recordings pre and post TFLLR application (mean ± SD, p=0.68, n = 12, 5 males, Wilcoxon matched-pairs signed rank test, two-tailed). (j) The fold change in mean event frequency for each color-coded recordings. (k) Amplitude histograms for pooled recordings measured before (black) and after (gray) astrocyte stimulation. (l) The fold change in mean event amplitude for each color-coded recording (mean ± SD). All recordings were performed using symmetrical trans-membrane chloride concentrations.

DOI: https://doi.org/10.7554/eLife.31629.012

The following figure supplements are available for figure 8:

**Figure supplement 1.** Western blot showing Par1 immunoreactivity in cortical lysates and similar Par1 levels in wild-type and MeCP2 null cortex.
DOI: https://doi.org/10.7554/eLife.31629.013

**Figure supplement 2.** The relative contributions of inward and outward currents following TFLLR activation in MeCP2 null barrel cortex.
DOI: https://doi.org/10.7554/eLife.31629.014

DMNP-EDTA caged calcium (4 mM DMNP-EDTA, 3.6 mM $CaCl_2$) for 10 m via the patch pipette. Calcium was then uncaged in the astrocyte via a 1 s LED light flash while the frequency of synaptic currents in the neighboring neuron was recorded (*Figure 11a*). To ensure that this uncaging protocol was effectively increasing intracellular calcium, we also bulk loaded slices with the calcium indicator Fluo4FF-AM. Photolysis in astrocytes effectively increased the somatic calcium levels for both wild-type ($\Delta F/F_0 = 0.18 \pm 0.06$, n = 6) and MeCP2 null animals ($\Delta F/F_0 = 0.26 \pm 0.05$, n = 4) (*Figure 11—figure supplement 1a–c*).

In wild-type slices, uncaging of calcium loaded DMNP-EDTA resulted in an average 1.39 ± 0.08 fold increase in the frequency of synaptic currents in the neighboring neuron (pre-flash: 250 ± 94 events/m, post-flash: 326 ± 75 events/m, p=0.0003, n = 16) (*Figure 11b,c*). In control experiments, photolysis of wild-type astrocytes dialyzed with an empty cage did not alter the frequency of synaptic currents (pre-flash: 109 ± 66 events/m, post-flash: 115 ± 75 events/m, n = 4) (*Figure 11—figure supplement 1d–f*). When uncaging was repeated using MeCP2 null slices, one of the 16 pairs tested responded a large increase in event frequency. However, overall, there was no significant change in the frequency of synaptic currents (pre-flash: 227 ± 112 events/m, post-flash: 237 ± 121 events/m, p=0.23, n = 16) (*Figure 11d–f*).

## Discussion

There is precedent from both electrophysiological recordings and calcium imaging for the ability of astrocyte stimulation to affect neuronal function in the brains of healthy mice (*Nedergaard, 1994*; *Kang et al., 1998*; *Fellin et al., 2004*; *Perea and Araque, 2005*; *Jourdain et al., 2007*; *Martín et al., 2015*; *Schipke et al., 2008*; *Benedetti et al., 2011*). The most commonly reported form of astrocyte-mediated modulation of neuronal signaling, also reflected in our study, involves changes in the frequency of neuronal synaptic currents mediated by the neurotransmitters GABA and glutamate (*Kang et al., 1998*; *Benedetti et al., 2011*; *Araque et al., 1998*; *Fiacco and McCarthy, 2004*; *Jourdain et al., 2007*; *Liu et al., 2004*). The increase in frequency for both neurotransmitters was not accompanied by increases in the average amplitudes, pointing to specific effects on presynaptic release and not postsynaptic receptors. GABA is of particular interest due to its dual role played in both excitation and inhibition of neuronal circuitry (*Newman, 2003*; *Yang et al., 2003*; *Jo et al., 2014*; *Zonta et al., 2003*). In our p10-12 postnatal recordings from layers 2/3 cortical neurons in wild-type mice, GABA is excitatory, consistent with previous studies performed at this age (*Ben-Ari, 2002*). Thus, in our neuronal recordings, GABAergic and glutamatergic transmission are both excitatory.

We show through wild-type astrocyte-neuron paired recordings, three different modes of astrocyte-specific stimulation that can effectively increase excitatory synaptic transmission. Both direct depolarization of the astrocyte using the patch pipette and calcium uncaging in the astrocyte represent non-physiological means of activation of both GABA and glutamatergic neuronal signaling. A physiological means of activating the astrocyte utilized local application of an astrocyte-specific agonist (TFLLR) of the metabotropic Protease-activated receptor 1 (Par1). Activation of the astrocyte via

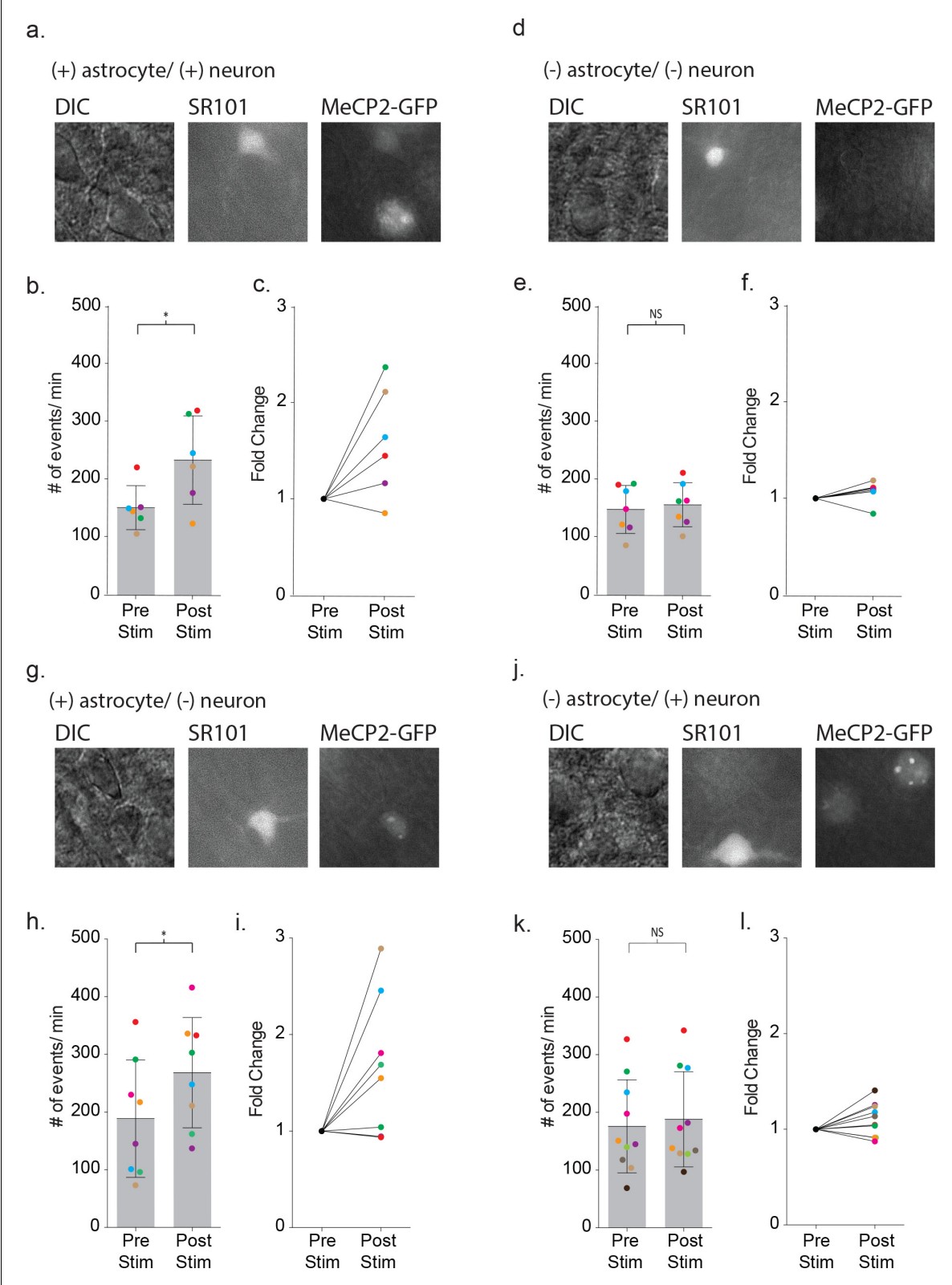

**Figure 9.** Defective signaling in astrocyte-neuron pairs results from the MeCP2 null astrocytes and not the neurons. (a) Images shown the DIC image (left), SR101 + astrocyte (middle) and GPF + astrocyte and GFP + neuron (right). (b) The mean ± SD event frequency for each color-coded neuron recording pre and post depolarization of the astrocyte (p=0.04, n = 6, 4 females, Paired t test, two-tailed). (c) The fold changes in event frequency for each color-coded pair following depolarization. (d–f) Data obtained from GFP - astrocytes and GFP - neuron color-coded recordings (p=0.24, n = 7, 5

*Figure 9 continued on next page*

*Figure 9 continued*

females, Paired t test, two-tailed). (g–i) Data obtained from GFP + astrocyte and GFP - neuron color-coded recordings (p=0.02, n = 8, 5 females, Paired t test, two-tailed). (j–l) Data obtained from GFP - astrocytes and GFP + neuron color-coded recordings (p=0.1176, n = 10, 6 females, Paired t test, two-tailed).

DOI: https://doi.org/10.7554/eLife.31629.015

Par1 works primarily through Gα signaling, leading to $IP_3$-mediated release of calcium through the endoplasmic reticulum. The Par1 receptor expresses at high levels in astrocytes (*Hermann et al., 2009*; *Park et al., 2015*; *Lee et al., 2007*), a finding confirmed by our immunohistochemistry on cortical astrocytes. There is precedent for Par1 receptor expression in certain neurons as well as evidence for their depolarization in response to TFLLR application. In the best studied example, a small fraction of dentate gyrus neurons responded to TFLLR with a depolarizing response, and only under conditions where bath application was very long (>120 s) compared to our study (*Han et al., 2011*). Evidence supporting a direct involvement of astrocyte Par1 receptors in neuronal activation comes from two of our findings. First, our immunohistochemistry only detected Par1 in the astrocytes and second the TFLLR-induced response was completely blocked when the astrocyte was dialyzed with the calcium buffer BAPTA. Finally, our findings in cortex, that Par1 in astrocytes stimulates adjacent neurons, are supported by a previous study in the nucleus of the solitary tract (*Vance et al., 2015*). Thus, we are confident that TFLLR was leading to cortical neuron excitability through actions on the astrocytes.

A central question that surrounds astrocyte effects on neuronal excitability is the identity of the factor/s released causal to the initiation of signaling. The list of published candidates includes glutamate (*Parpura et al., 1994*), ATP (*Newman, 2003*), D-serine (*Yang et al., 2003*), GABA (*Jo et al., 2014*), and derivatives of arachidonic acid (prostaglandins, epoxyeicosatrienoic acid, and 20-hydroxyeicosatetraenoic acid) (*Zonta et al., 2003*). For our studies, glutamate and GABA specifically were tested because of their documented roles as gliotransmitters in barrel cortex (*Min and Nevian, 2012*; *Benedetti et al., 2011*). Through alteration of the trans-membrane chloride concentration gradient we created outwardly (GABA) and inwardly (glutamate) directed synaptic events to test whether either GABA or glutamate receptor blockade would shut down both classes of synaptic currents in response to TFLLR. Application of GABA antagonists resulted in inhibition of outward currents but the activation of inward currents remained, thus eliminating $GABA_A R$ involvement in the initiation of the circuitry. By contrast, applying glutamate receptor antagonists blocked the TFLLR induced increase in the majority of outward GABA mediated responses and most, but not all of the glutamate mediated inward responses. The persistence of inward responses could reflect an incomplete block of glutamate receptors or involvement of additional unidentified chemical activators. As a further test for involvement of a direct glutamate receptor response we performed time-locked neuronal recordings in response to astrocytic depolarization. Examination of the pooled event timing revealed a significant, but delayed, increase in frequency within 40 to 60 ms of astrocyte depolarization. These responses were further tested for evidence of direct activation through inclusion of TTX which blocks neuronal mediated effects. TTX eliminated the increase in event frequency associated with the depolarization pointing to release from neurons and not astrocytes. Therefore, the putative molecule/s underlying astrocyte modulation of neuronal signaling in our context is still at large.

Our study was successful in assigning a functional requirement for MeCP2 expression to a specific cell type. Recent studies have drawn into question the exclusive role of neurons in the progression of RTT, a disease resulting from loss of MeCP2 expression in both cell types. For example, the progression of overt RTT-like behaviors in mice is blunted following postnatal restoration of MeCP2 just in astrocytes (*Lioy et al., 2011*), consistent with reversibility of the disease with ubiquitous MeCP2 expression (*Guy et al., 2007*). Additionally, the astrocytes in a RTT mouse have reduced sensitivity in their detection of carbon dioxide in CNS respiratory centers (*Turovsky et al., 2015*) and the astrocytic expression of MeCP2 is critical in maintaining adaptive responses to respiratory $CO_2$ chemosensitivity (*Garg et al., 2015*). Our initial recordings from globally MeCP2 null cortex could have been explained by defects in either astrocytes or neurons, or both cell types. However, owing to our ability to unequivocally stimulate individual astrocytes by means of depolarization, we were able to test all combinations of neurons and astrocytes. For this purpose we utilized the female mosaic

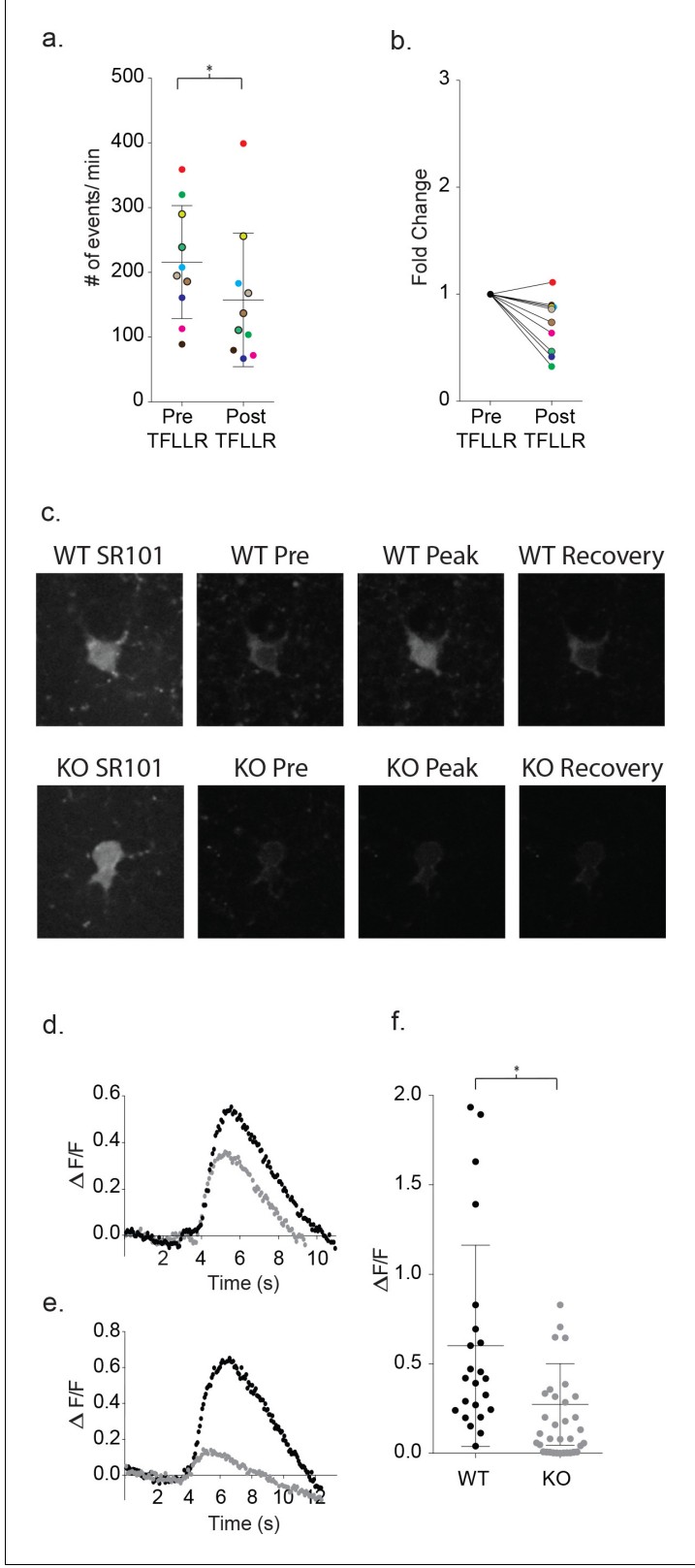

**Figure 10.** Increased astrocyte calcium is required for neuronal signaling and is blunted in Mecp2 deficient astrocytes. (a) Clamping calcium to low intracellular levels prevents the TFLLR astrocyte-mediated increase in neuronal event frequency. Six of the recordings were made in 10 mM free BAPTA recordings and four in the calcium BAPTA buffer (black outlined circles). The mean event frequency from each color-coded recording, pre-

*Figure 10 continued on next page*

*Figure 10 continued*

and post-application of 500 μM TFLLR (p=0.02, n = 10, 2 males and 2 females, Wilcoxon matched-pairs signed rank test, two-tailed). (b) The fold change for each of the color-coded recordings. (c) Images showing SR101 labeled wild-type (top; WT) and Mecp2 null (lower; KO) astrocytes loaded with the calcium indicator fluo4FF-AM. Selected images of calcium responses corresponding to before the peak response and after recovery from 500 μM TFLLR application. (d, e) Two sample data sets showing $\Delta F/F_0$ measurements from wild-type astrocytes (black) and MeCP2 null astrocytes (gray) during a round of TFLLR application. The top sample reflects one of the strongest null responders and the bottom sample is representative of the overall mean. The time courses for wild-type and null recordings are similar and the dip below baseline following recovery reflects the time-dependent bleaching of the calcium indicator. (f) The scatterplot of $\Delta F/F_0$ for individual recordings comparing wild-type (mean $\Delta F/F_0$ = 0.60 ± 0.56, n = 23, 5 males and 4 females) and MeCP2 null recordings (mean $\Delta F/F_0$ = 0.27 ± 0.23, n = 23, 8 males; p=0.012, Mann Whitney test, two-tailed). Only those MeCP2 null astrocytes that responded to TFLLR were included in the statistical analysis. Mean +/- SD for the overall data are indicated.

DOI: https://doi.org/10.7554/eLife.31629.016

**Figure 11.** Uncaging intracellular calcium does not rescue astrocyte-neuron signaling. (a) Sample recordings (3 s) from a neighboring cortical neuron before (upper) and after (lower) flash uncaging of intracellular calcium in the astrocyte. (b) The scatterplot of pre- and post-flash color-coded responses showing the mean ± SD (p=0.0003, n = 16, 3 males and two female, Wilcoxon matched-pairs signed rank test, two-tailed). (c) The associated fold change in event frequency for each of the color-coded recordings. (d–f) Measurements made from MeCP2 null neurons when MeCP2 null astrocytes were loaded with caged calcium showing, in all but one recording, no significant increase in neuronal event frequency upon uncaging of calcium (p=0.23, n = 16, four males Wilcoxon matched-pairs signed rank test, two-tailed).

DOI: https://doi.org/10.7554/eLife.31629.017

The following figure supplement is available for figure 11:

**Figure supplement 1.** Fluo4FF-AM calcium imaging of astrocytic somatic calcium signals ($\Delta F/F_0$) from photolysis of DMNP-EDTA caged calcium in wild-type and MeCP2 null slices.

DOI: https://doi.org/10.7554/eLife.31629.018

MeCP2 brain, which reflects the same condition as in human females with RTT. Our findings showed that MeCP2 expression in astrocytes was required to restore a normal neuronal synaptic response to astrocyte stimulation, even when the neurons lacked MeCP2. This indicates that in the female RTT brain, only in the case where there is a wild-type astrocyte coupled to a neuron, will there be normal astrocyte-neuronal physiology. As in neurons,~50% of the astrocytes in the mosaic female will usually be defective. Our results therefore point to a strong contribution by astrocytes to the abnormal brain circuitry underlying RTT.

What defects imparted by MeCP2 deficiency in astrocytes account for blunted neuronal signaling? Calcium signaling pathways are often involved in gliotransmitter release (*Bazargani and Attwell, 2016*) and both our ability to inhibit neuronal signaling through inclusion of a strong calcium buffer in the astrocyte as well as activation through calcium uncaging point to involvement of calcium (*Kang et al., 1998*; *Benedetti et al., 2011*; *Bal-Price et al., 2002*; *Lee et al., 2011*). Also, TFFLR acts through the Par1 receptor to release calcium from intracellular stores (*Junge et al., 2004*; *Hollenberg et al., 1997*; *Ubl and Reiser, 1997*). Less certain however are the links between astrocyte depolarization and potential changes in intracellular calcium. Both T type and L type voltage dependent calcium channels have been identified in acutely dissociated astrocytes (*Latour et al., 2003*; *Verkhratsky et al., 1998*). The extent to which such channels are expressed throughout the astrocyte population is not known and several studies have contested their expression in astrocytes. We were not able to identify voltage dependent calcium currents or increases in intracellular calcium using a depolarization protocol, which leaves the door open to mechanistic interpretations.

Although we couldn't identify with certainty a depolarization-mediated mechanism, we tested for a potentially unifying role of the calcium response. This test was called for by our findings that dialysis of the astrocyte with BAPTA blocked neuronal signaling and the blunted TFLLR-mediated calcium response in RTT astrocytes. However, our attempts to rescue defective signaling by direct calcium uncaging in RTT astrocytes failed in all but a single recording. From this result we surmise that the mechanisms in calcium handling that are associated with MeCP2 deficiency in astrocytes are not solely responsible for the reduced astrocyte-mediated stimulation of neurons. Given that loss of MeCP2 results in global gene expression changes in the brain, some of which have been linked to astrocytic functions that could impact signal transduction, such as microtubule dynamics and vesicle trafficking (*Delépine et al., 2016*), this result is not completely unexpected and underscores the complexity of alterations with the loss of MeCP2.

Taken together, our physiology reveals a perturbation in synaptic communication in mouse models of RTT, manifesting as decreased astrocyte neuronal excitatory signaling. The decreased signaling can account, at least in part, for the compromised ability of RTT mouse models to perform normal behavioral tasks, as well as for amelioration of RTT-like phenotypes after restoration of MeCP2 in astrocytes. In addition to reduced overall excitatory neuronal signaling, there are potential implications reflected in an altered excitatory-inhibitory (E/I) synaptic balance reported in RTT (*Feldman et al., 2016*). Normally, activation of the GABA$_A$R switches from being excitatory to inhibitory during early postnatal mouse development (*Araque et al., 1998*; *Ben-Ari et al., 1989*; *Obata et al., 1978*; *Owens et al., 1996*; *Dammerman et al., 2000*). This switch results from the expression of the chloride exporter KCC2, which establishes the asymmetrical trans-membrane concentration gradient (*Blaesse et al., 2009*; *Ganguly et al., 2001*). In both humans with RTT and mouse models, a delay in the expression of the chloride exporter occurs, thus causing developmental delay in the switch to inhibition (*Duarte et al., 2013*; *Tang et al., 2016*; *Feldman et al., 2016*). Consequently, the absence of astrocytic contributions to neuronal signaling in RTT places regions of the brain, which are critically dependent on E/I balance, at greater risk.

## Materials and methods

### Animals

All animal procedures were approved by the Oregon Health and Science University Institutional Animal Care and Use Committee under protocol number IP00000284. Mice were kept on 12 hr light/dark cycles and were housed with their littermates. Three genotypes were used in this study: C57BL/6J, *Mecp2tm1.1Bird* (*Mecp2/y*; catalog 003890; [*Guy et al., 2001*]), and *Mecp2tm3.1Bird* (*Mecp2EGFP*;

catalog number 014610; [*Lyst et al., 2013*]). All mice were obtained from Jackson Laboratory and crosses were maintained on a C57BL/6J background. Mice were genotyped by PCR as described previously (*Lioy et al., 2011*; *Lyst et al., 2013*). To generate heterozygous $Mecp2^{Bnull/EGFP}$ females (mice carrying a *Mecp2-EGFP* allele and a germ-line *Mecp2*-null mutation), $Mecp2^{EGFP/y}$ mice were crossed to heterozygous $Mecp2^{Bnull/+}$ mice.

## Brain slice preparation

Mice between ages p10 and p12 were anesthetized with isoflurane prior to decapitation and removal of the brain. Coronal slices (200 μm) were cut from the barrel cortex of the right hemisphere using a vibratome (Leica VT1200s). An ice-cold cutting solution contained: (in mM) 110 CholineCl, 2.5 KCl, 1.25 $NaH_2PO_4$, 1.3 Na-ascorbate, 3 Na-pyruvate, 10 glucose, 25 $NaHCO_3$, 0.5 $CaCl_2$, 7 $MgCl_2$ and adjusted to pH 7.3. Slices were then placed in a recovery solution for 35 m at 35°C which contained: (in mM) 119 NaCl, 2.5 KCl, 1 $NaH_2PO_4$, 1.3 Na-ascorbate, 3 Na-pyruvate, 10 glucose, 26 $NaHCO_3$, 1.5 $CaCl_2$, 1 $MgCl_2$ and 100 nM SR101 to label astrocytes. The recovery solution was continuously bubbled with Carbogen (Carbon dioxide 5%, balanced oxygen). Slices were then transferred to a lower temperature recovery solution (23°C) for an additional 35 m and used within 5 hr.

## Electrophysiological recordings, analyses and statistics

All recordings were performed at ~32°C from layer 2/3 barrel cortex astrocytes and neurons. Astrocytes and neurons were visualized by means of an IR-DIC 40x W achroplan objective using a Zeiss Axiocam MRm (Zeiss, Oberkochen, Germany). Astrocytes were identified on the basis of Sulforhodamine 101 red fluorescence (Sigma-Aldrich, St. Louis, MO). In MeCP2-GFP knock-in mice, expression of MeCP2 was identified on the basis of green fluorescence.

The extracellular recording solution contained: (in mM) 110 NaCl, 2.5 KCl, 1 $NaH_2PO_4$, 10 glucose, 26 $NaHCO_3$, 2 $CaCl_2$, and 1.3 $MgCl_2$. The recording solution was continuously bubbled with Carbogen to maintain the pH at 7.3. The internal solution for symmetrical chloride recordings contained: (in mM) 140 CsCl, 10 Cs-HEPES, 10 Cs-EGTA, pH 7.35. The internal solution for low intracellular chloride contained: (in mM) 130 K-Gluconate, 10 KCl, 10 K-EGTA, 10 K-HEPES, pH 7.38. For test for a calcium requirement in signaling the patch pipette contained either 10 mM BAPTA (0 free calcium) or 10 mM BAPTA with 4.83 mM $CaCl_2$ (100 nM free calcium). All recordings utilized a dual headstage EPC10 HEKA amplifier. Data was acquired using Patchmaster software (HEKA elektronik, Lambrechy/Pfalz, Germany) at a sampling rate of 10 kHz and Bessel filtered offline at 500 Hz.

For perforated patch neuron recordings, the tip of the patch pipette was filled with gramicidin-free internal solution prior to backfilling with internal solution containing 100 μg/mL gramicidin (Calbiochem, San Diego, CA). Following seal formation the resistance was monitored until it dropped to <50 MΩ. Establishment of the whole cell recording was then confirmed on the basis of a large voltage activated sodium current.

The amplitude and frequency of individual synaptic currents were determined using a combination of Patchmaster, ClampFit (Axon Instruments, Sunnyvale, CA) and Mini-analysis software (Synaptosoft, Decatur, GA). After 500 Hz filtering the detection threshold was set to 3x RMS noise for symmetrical chloride recordings and to 2x RMS for low chloride internal solution recordings. Statistical analyses were carried out using GraphPad Prism6 software (La Jolla, CA). The paired t-tests used either a parametric paired-t test or the Wilcoxon matched-pairs signed rank test and the unpaired t-test used the Mann-Whitney test. All t-tests were two-tailed and a confidence level of 95% was used. Statistical significance was based on $p < 0.05$.

The agonists L-glutamic acid, TFLLR-NH2, and GABA were pressure applied through a patch electrode via a picospritzer (Parker Instruments, Boaz, Al). The antagonists gabazine (SR95531 hydrobromide), DL-AP5, and NBQX were incubated with the slice for minimum of 10 m included in puffer pipette along with the appropriate agonist. All agonists and antagonists were obtained from Tocris (Bristol, UK). Tetrodotoxin (TTX) was obtained from Alomone Labs (Jerusalem, Israel)

## Calcium imaging and calcium uncaging

Astrocytic somatic calcium signals were quantified using 10 μM Fluo4FF-AM (Molecular Probes, Eugene, OR). After recovery at 35°C, slices were incubated in extracellular recording solution containing equal volumes of calcium indicator and 20% (w/v) Pluronic acid (Molecular Probes, Eugene,

OR). These slices were maintained at 23°C for 25 m during which they were shielded from light and continuously bubbled with Carbogen. Slices were then transferred to the microscope stage and washed for 30 additional minutes with bubbled extracellular recording solution to allow for de-ester-ification of the calcium indicator. Live calcium signals were acquired using a Yokogawa CSU-X1 con-focal spinning disc equipped with a Hamamatsu EM-CCD digital camera (Imagem-x2). A 488 nm laser line was used for calcium signals and 561 nm line was used for SR101 fluorescence (OBIS Coherent, Santa Clara, CA). Laser power was set to <50%, which corresponded to a final output of 2.3 mW for the 488 nm line and 1.2 mW for the 561 nm line. Images were captured every 75 ms beginning 2 s before drug application and continued until the calcium signal decayed. Micro-man-ager software (UCSF, San Francisco, CA) was used to acquire images and Image J (NIH, Bethesda, MD) and Microsoft Excel (Redmond, WA) were used for offline analysis. Throughout all of the experi-ments involving calcium imaging the laser and camera gains were held constant. Thus, it became necessary to exclude those astrocytes, both mutant and wild-type, that had high resting levels of cal-cium in order to prevent signal saturation. To quantitate calcium signals in astrocytes that qualified for analysis, the entire somatic region was selected as a region of interest (ROI). Then, for each ROI, the time dependent changes in signal intensity were quantitated as $\Delta F/F_0$ as follows. First, the aver-age pixel intensity within each ROI was corrected for background fluorescence. To establish true background levels the average pixel intensity was determined using cells not incubated with the cal-cium indicator. Second, the background corrected basal fluorescence signal intensity ($F_0$) was deter-mined on the basis of the average pixel intensity measured for the ROI during the two seconds prior to drug application. Third, the change in average fluorescence intensity for the ROI ($\Delta F$) was deter-mined by subtracting the average pixel intensity for the ROI by the background-subtracted basal fluorescence ($F_0$).

For calcium uncaging experiments, 4 mM DMNP-EDTA (Molecular Probes, Eugene, OR) was loaded with 3.6 mM $CaCl_2$ for 10 m in the dark. The astrocyte was then dialyzed with the loaded cage via the patch pipette along with 0.1 mM Alexa Fluor 488 for at least 10 m. A shuttered, exter-nally mounted fiber optic was used to trigger a 1 s flash from an X-Cite 120 EXFO LED. Control astrocytes were dialyzed with an empty DMNP-EDTA cage and Alexa Fluor 488.

For certain experiments, the efficacy of the uncaging protocol was validated by use of a calcium indicating dye during the uncaging. For this purpose slices were incubated with 10 μM Fluo4FF-AM and astrocytes were loaded with 4 mM DMNP-EDTA caged calcium as previously described. Resting calcium levels were measured for ~2 s using the 488 nm laser line after which the calcium was unc-aged using a 1 s flash. At the time of uncaging, it was necessary to close the light path to the cam-era, limiting the calcium imaging and analysis to the time following termination of the flash.

## Immunohistochemistry

Coronal 80 μm slices were cut from barrel cortex in ice cold 1x PBS, the cortex was removed and transferred to 4% paraformaldehyde (PFA) and fixed for 3 hr at 4°C. Slices were washed with 1x PBS containing 20 mM glycine. Slices were then permeabilized with 1x PBS containing 0.2% triton x-100 for 15 m at 23°C. Slices were then blocked with 1x PBS containing 3% BSA and 5% normal goat serum for 30 m at 23°C followed with incubation in the respective primary antibody overnight at 4°C in the PBS-based blocking solution. After overnight incubation, the slices were washed and then incubated in secondary antibody for 2 hr at 23°C in the PBS-based blocking solution. Following exposure to the secondary antibody, slices were washed and then mounted using Prolong gold with DAPI. Primary antibodies used were: NeuN (1:200, Neuronal nuclei clone A60, MAB377, mouse monoclonal, Millipore, Darmstadt, Germany), GFAP (1:500, Glial Fibrillary Acidic Protein, Z0334, rab-bit polyclonal, Dako, Santa Clara, CA), Par1 (1:100, Thrombin R, S-19, SC-8204, Goat polyclonal, Santa Cruz Biotechnology, Dallas, TX). Secondary antibodies used were: Alexa Fluor donkey anti goat-568 (1:500, A-11057), Alexa Fluor donkey anti rabbit-647 (1:500, A-31573), Alexa Fluor donkey anti mouse-488 (1:500, A-21202). All secondary antibodies were purchased from Thermo Fisher (Waltham, MA).

## Western blotting

Coronal 80 μm slices were cut from barrel cortex in ice cold 1x PBS, the cortex was removed and immediately lysed in RIPA buffer (25 mM Tris ph 7.6), 150 mM NaCl, 1% NP40, 1% deoxycholate,

0.1%SDS, benzoate nuclease, protease inhibitor). Protein concentration was determined via BCA assay. 40 μg of protein (boiled with loading dye at 70°C for 10 m) was run on a 4–12% bis tris gel with MOPS at 200 V for 1 hr. Gel was transferred to a nitrocellulose membrane using a methanol-based buffer for 1 hr at 100 V. Blot was blocked in 1x TBST with 3% BSA for 1 hr at 23°C and then incubated in primary antibody with 1x TBST and 1.5% BSA over night at 4°C. Following primary incubation, the blot was washed and then incubated in secondary antibody for 1 hr in 1xTBST with 3% BSA at 23°C. Blot was then washed before imaging on ODYSSEY. Primary antibodies used were: Par1 (1:5000, Thrombin R, S-19, SC-8204, Goat polyclonal, Santa Cruz Biotechnology, Dallas, TX), α-tubulin (1:10000, mouse monoclonal, DSHB #AA4.3, Iowa City, Iowa), MeCP2 (1:5000, rabbit monoclonal, cell signaling D4F3, 3456 s, Danvers, MA). Secondary antibodies were conjugated to IR-fluorophore for imaging on Li-Cor Odyssey machine: goat anti mouse- IR680 (Thermo Fisher 35518, Waltham, MA), goat anti rabit-IR680 (Thermo Fisher #33568, Waltham, MA) donkey anti goat- IR 800 (Li-Cor 925–32214, Lincoln, NE).

## Additional information

### Competing interests

Gail Mandel: Reviewing editor, *eLife*. The other authors declare that no competing interests exist.

### Funding

| Funder | Grant reference number | Author |
|---|---|---|
| National Institutes of Health | HD081037 | Paul Brehm<br>Gail Mandel |
| Rett Syndrome Research Trust | RSRT | Gail Mandel |

The funders had no role in study design, data collection and interpretation, or the decision to submit the work for publication.

### Author contributions

Benjamin Rakela, Conceptualization, Data curation, Formal analysis, Validation, Investigation, Visualization, Methodology, Writing—original draft, Writing—review and editing; Paul Brehm, Conceptualization, Resources, Data curation, Formal analysis, Supervision, Funding acquisition, Validation, Visualization, Methodology, Writing—original draft, Project administration, Writing—review and editing; Gail Mandel, Conceptualization, Resources, Data curation, Formal analysis, Supervision, Funding acquisition, Validation, Investigation, Visualization, Methodology, Writing—original draft, Project administration, Writing—review and editing

### Author ORCIDs

Benjamin Rakela (iD) https://orcid.org/0000-0002-4264-0562
Gail Mandel (iD) https://orcid.org/0000-0003-2085-4516

### Ethics

Animal experimentation: All animal procedures were approved by the Oregon Health and Science University Institutional Animal Care and Use Committee under protocol number IP00000284.

### Decision letter and Author response

Decision letter https://doi.org/10.7554/eLife.31629.021
Author response https://doi.org/10.7554/eLife.31629.022

## Additional files

### Supplementary files

• Transparent reporting form

DOI: https://doi.org/10.7554/eLife.31629.019

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
