## [Decision Letter]

Thank you for submitting your article "Astrocytic modulation of excitatory synaptic signaling in a mouse model of Rett syndrome" for consideration by *eLife*. Your article has been favorably evaluated by Richard Aldrich (Senior Editor) and three reviewers, one of whom is a member of our Board of Reviewing Editors. The reviewers have opted to remain anonymous.

The reviewers have discussed the reviews with one another and the Reviewing Editor has drafted this decision to help you prepare a revised submission.

Summary:

This is an interesting study that investigates the impact of MeCP2 and astrocyte gating of neuronal activity in normal and mouse model of Rett syndrome. The authors employ a range of techniques to depolarize astrocytes and measure the frequency and amplitude of events on a neighboring neuron. First, the authors were able to induce the increase in the frequency, not amplitude, of events on neighboring neurons utilizing three techniques (intracellular depolarization, astrocytic-specific ligand, and calcium uncaging) to activate astrocytes. Second, this effect was not directly related to GABA or glutamate release/signaling. Thirdly and most interestingly, the persistent increase was dependent on astrocytic, not neuronal, expression of MeCP2.

Overall, the data are compelling and the topic is interesting. Understanding the contribution of astrocytes to neuronal and circuit dysfunction is important. The mosaic model is a powerful way to test cell-type dependence of the phenotype. However, there were also a number of questions and revisions raised by reviewers that should be addressed as outlined below, including the addition of statistics, p values, experimental details to the key figure and results. The authors show that upon non-physiological depolarization of astrocytes, it can impact synaptic activity in surrounding neurons but fail to identify a physiologically-relevant stimuli or potential mechanism. If this is challenging to address experimentally in a timely manner, it is important that the authors include critical and thoughtful discussion of these issues in the revised paper. In addition, some consolidation of Results section and figures is needed to increase clarity and readability for non-experts.

*Reviewer #1:*

The manuscript from Rakela and colleagues investigates the impact of MeCP2 and astrocyte gating of neuronal activity in normal and mouse model of Rett syndrome. In this study, the authors employ a range of techniques to depolarize astrocytes and measure the frequency and amplitude of events on a neighboring neuron. First, the authors were able to induce the increase in the frequency, not amplitude, of events on neighboring neurons utilizing three techniques (intracellular depolarization, astrocytic-specific ligand, and calcium uncaging) to activate astrocytes. Second, this effect was not directly related to GABA or glutamate release/signaling. Thirdly and most interestingly, the persistent increase was dependent on astrocytic, not neuronal, expression of MeCP2. I agree with the authors that their work to discriminate the expression of MeCP2 as necessary across the astrocyte/neuronal interaction is important and clever in its execution. While the authors were able to show that GABA/Glutamate signaling was not necessary for persistent changes, the lack of mechanistic insight leaves many questions that should be addressed before publication:

1) As indicated in the discussion, Lioy et al. (2011) restored many core features (both behaviorally and anatomically) in a MeCP2-deficient mouse with the selective expression of MeCP2 in only astrocytes. In the study, they showed an increase in the number of vGluT1 terminals and restored morphology. Does the persistent effect shown by Rakela et al. depend on the increase in presynaptic release of vGluT1 synapses? If persistently altered for long periods of time, would it be capable to capture either a morphological or histological change in terminals?

2) On a related note, what characteristics of presynaptic release are altered in the face of astrocytic gating? Does the number of release sites increase? The lack of amplitude changes suggests that quantal size is not changing, but how do these changes impact short-term plasticity measures?

*Reviewer #2:*

This is an interesting study that reports how astrocyte stimulation affects synaptic response onto pyramidal neurons in control mice and in Rett mouse models. Overall, the data are clear and the topic is interesting and important. I have several suggestions that the authors could accommodate with some revisions.

1) For the Par-1 immuno staining, please present some analyses for the expression within astrocytes. Include some average data. The Par-1 response is an important stimulus that is used in this paper, but the reader needs to be convinced that Par-1 receptors exist in cortical astrocytes.

2) Please provide a clear reference to a published paper showing that gramicidin pores are not permeable to chloride. Again, changing the Cl^-^ reversal potential is a nice approach used in this study, but the reader needs to be convinced that gramicidin perforated patches behave in the way the authors state. This knowledge is critical to interpret the result in Figure 4 that GABA is excitatory at this stage of development in the cortex. If other people have addressed this topic before then additional references should be provided.

3) The presentation of the results is a bit long-winded and ponderous. Perhaps sharpen the text to make it briefer and more to the point.

4) The authors make the point clearly that they do not know the identity of the substance released from astrocytes to trigger the altered neuronal signaling. If they are sure it is not glutamate release, then they should state this explicitly. Also, could the substance be a change in ATP, K^+^ and or Ca^+^ in the extracellular fluid? If so, please discuss.

5) It is understood that tools to manipulate astrocytes are limited and the authors have had to use strong depolarization, Par-1 agonists and Ca^2+^ uncaging. As experimental tools these are fine, but the authors should guide the readership by stating in the Discussion that it is likely that all three of these methods are non-physiological and that the conclusions should be interpreted with these thoughts in mind. Thus it is hard to imagine a scenario when astrocytes depolarize to +60mV.

6) Add statistics, p values, n numbers and statistical tests to the key figures so the reader can see what is happening just by looking at the figures.

7) Many of the scatter graphs show mean +/- sd. This is fine, but I wonder if it would be better to show box and whisker plots with mean, sd and sem? I suspect some readers will appreciate such graphs more than mean +/- sd.

Overall, this study is a valuable addition to the literature and another round of hard work to tighten up the aforementioned aspects will make the study better. I can't comment too much on the mouse model of Rett itself as this is beyond my expertise. However, this aspect will hopefully be addressed by one of the other reviewers.

*Reviewer #3:*

This paper examines changes in astrocyte-driven effects on neurotransmission in a mouse model of Rett syndrome. The authors suggest that a mechanism dependent on calcium dependent astrocyte signaling increases the frequency of spontaneous EPSCs and IPSCs. This form of gliotransmission is absent in MeCP2 deficient (male) mice and shows rescue in WT astrocytes in female mosaic mice. The strength of the paper is in the description of the astrocytic locus of the phenotype. The mosaic model is a powerful way to test cell-type dependence of the phenotype. The weaknesses of the paper revolved around the lack of mechanistic insight and the overinterpretation of experiments.

Figure 2 and histograms elsewhere. These were apparently built from the data from >1 cell, and according to the Results (subsection “Astrocytes mediate increases in cortical neuron signaling”, second paragraph), event frequency varied greatly among cells. How were histograms built to avoid overburdening the distribution with events from high-frequency cells and to thus keep the amplitude distribution representative?

Figure 2. What is the interpretation of the increase in select large events? The statistical treatment of this is unclear.

Regardless of the pulse amplitude used for depolarizing stimulation, the physiological relevance of the strong depolarization of astrocytes is highly questionable. It is good that the investigators use a second method of local TFLLR application, but the relevance and specificity of this are also questionable (see below). Were other physiologically-based methods of astrocyte stimulation attempted? Were they unsuccessful?

Par1 activation with TFLLR occurs in isolated neurons and causes them to fire action potentials (PMID 21827709). Thus, the possibility of a direct TFLLR on neurons needs to be treated with respect.

The Ca^2+^ signal initiated by astrocyte depolarization should be documented in the authors' hands, using the brief-pulse protocols used to elicit the PSC frequency increase. A few papers are cited, but voltage-gated Ca^2+^ influx in astrocytes is not universally observed.

The authors seem to want to interpret experiments in Figure 4–Figure 8 as a test of the astrocyte derived signal (e.g. Discussion, second paragraph). However, the experiments testing the nature of gliotransmitter (glutamate, GABA) are confounded by the same substances serving as neurotransmitters. A likely model to explain the data is that substance X, released from astrocytes, increases spiking selectively in principal neurons surrounding the recorded cell (perhaps because only principal cells express the receptor for X). This accounts for the increase in sEPSCs. Local synaptic interactions between principal neurons and interneurons then explain the increase in sIPSCs. When viewed through the lens of this explanation, the experiments designed to determine the polarity of GABA signaling seem tangential.

The attempts to measure EPSCs and IPSCs by clamping midway between reversal potentials appear fraught. Membrane noise is high in Figure 5, likely because of the depolarized Vm value. This will increase false positives and perhaps false negatives. The false positive problem is highlighted by incomplete antagonism in Figure 5. Overall, it is difficult to believe the quantifications from this protocol.

Figure 8. Sample traces should be shown here. I have trouble reconciling the time course data with Figure 2. What am I missing? The statistical treatment here (2 peaks/) is not clear. Further, the basis for the time course data informing the likelihood of glutamate as mediator (subsection “Interdependence of GABA and glutamatergic signaling pathways”, last paragraph) are lost on me.

Figure 12: Raw images that serve as the basis for the Ca^2+^ imaging should be shown. How confident can the authors be that images derive from astrocytes? Wouldn't signals from astrocyte processes be most relevant to the studies? How were signals from neurons/neuron dendrites excluded?

Supplemental Figure 5. Ca^2+^ signals from neurons and astrocytes are shown, but the authors state that Fluo-AM preferentially labels astrocytes.

The results seem to suggest that both Ca increase/handling and downstream signal transmission are altered in MeCP2 astrocytes. This is an interesting conclusion but leaves one hanging, without more mechanistic explanation.

Other points:

Sex of WT mice in initial experiments should be specified.

A general description of statistical treatment is given in the Materials and methods. For increased transparency, I would like to see the statistical tests specified in the figure legends for each experiment.

Abstract, and perhaps elsewhere: 'astrocyte activation' has a specific connotation that should be avoided in this paper.

Subsection “Astrocytes mediate increases in cortical neuron signaling”, second paragraph. Why mention the decay information as data not shown? If this is important to the arguments presented, show the data. If it is not important, omit the reference.

Subsection “Astrocytes mediate increases in cortical neuron signaling”, second paragraph. The Results describe astrocyte stimulation as a pulse to +60 mV; the figure legend cites +140 mV.

[Editors' note: further revisions were requested prior to acceptance, as described below.]

Thank you for resubmitting your work entitled "Astrocytic modulation of excitatory synaptic signaling in a mouse model of Rett syndrome" for further consideration at *eLife*. Your revised article has been favorably evaluated by Richard Aldrich (Senior Editor), a Reviewing Editor, and 1 reviewer.

The manuscript has been improved, rewritten and streamlined as suggested. However, there are a few remaining issues that need to be addressed before acceptance, as outlined below.

1) Please clarify and explain rationale for the GABA experiments in Figure 5 and clarify the time course data in Figure 7.

2) Please address remaining concerns related to the design and interpretation of experiments in Figure 6 that suggest GABA/Glutamate signaling are not involved in the signaling cascade. Reviewer 3 raises two specific concerns related to Figure 6—figure supplement 11 and Figure 6 and suggests including raw traces and to help build confidence in the interpretation as summarized below.

*Reviewer #3:*

The authors have achieved a more effective and streamlined presentation in this revision. As noted by previous reviews, the main limitation remains the limited mechanistic insight. In addition, some important elements remain unclear.

The rationale for the GABA experiments in Figure 5 and Figure 6 seems to be that because GABAA receptor activation is excitatory at this age (Figure 5), GABA/GABAARs are candidates to mediate depolarization of the network and thus the increased sPSC frequency. The rationale and where these experiments are headed should be more clearly explained near the beginning of the description of Figure 5.

I cannot overcome my problems with the design of Figure 6 and thus the interpretations of experiments meant to exclude GABA/glutamate as participants in the signaling cascade. The authors seem to concede that S/N is a problem. This does not lend confidence to the interpretation of either positive or negative results in the figure. As it stands, the lack of full pharmacological block in Figure 6—figure supplement 1 suggests that false positives are a problem. Even more concerning is the high frequency of inward sPSCs in Figure 6 in the presence of NBQX/APV. Raw traces post antagonist may help build confidence. It seems to me that the proper design would have been to use CsCl pipet solution, -70 mV, and pharmacologically isolate GABAA sPSCs and in turn AMPAR sPSCs. This would not have affected GABA's influence on the network.

I am still confused about the time course data shown in Figure 7. Here the effect of depolarization fades by 150 ms, while in Figure 1 and Figure 4 the effect remains for 20-30 s.

---

## [Author Response]

Overall, the data are compelling and the topic is interesting. Understanding the contribution of astrocytes to neuronal and circuit dysfunction is important. The mosaic model is a powerful way to test cell-type dependence of the phenotype. However, there were also a number of questions and revisions raised by reviewers that should be addressed as outlined below, including the addition of statistics, p values, experimental details to the key figure and results. The authors show that upon non-physiological depolarization of astrocytes, it can impact synaptic activity in surrounding neurons but fail to identify a physiologically-relevant stimuli or potential mechanism. If this is challenging to address experimentally in a timely manner, it is important that the authors include critical and thoughtful discussion of these issues in the revised paper. In addition, some consolidation of Results section and figures is needed to increase clarity and readability for non-experts.

We reduced the number of figure supplements from 7 to 5, added the requested immunohistochemistry figure and further consolidated the primary figures from 13 to 11. We color-coded all recordings, in all of the appropriate figures, to provide easy visual links between the histograms showing the number of events and the corresponding fold changes. We now include statistics and gender within the figure legends for all experimental data. We also shortened and sharpened the Results section. We rewrote the paper to better describe the different forms of stimulation, pointing out directly that depolarization is likely non-physiological and, like so many other studied features of astrocytes, obscures mechanistic insights. We also point out, however, that without our use of depolarization, which acts on one specific astrocyte, we would not have been able to perform the novel experiments in the female mosaic brain (Figure 9) or the time-locked measurements that allowed us to test direct release of excitatory factors from the astrocyte (Figure 7). The female is the gender appropriate model for Rett syndrome, but most molecular and physiology studies have not exploited a clean way to differentiate the normal and mutant cells in the intact mosaic brain, relying instead on the hemizygous males. Regarding the issue of mechanisms underlying the depolarization-induced signaling, we tested for voltage dependent calcium channels as well as monitoring calcium levels during depolarization. We did not add these tests to the Results section because the results were all negative, leaving readers to ponder, and one reviewer had noted that our Results section was already “longwinded and ponderous”. Finally, we spelled out more clearly the actions of the agonist TFLLR. We maintain, based on many previous studies, that TFLLR activates a physiologically relevant pathway, taking issue with the idea that we rely solely on a non-physiological stimulus. Below we address each of the three reviewers’ concerns point by point.

Reviewer #1:[…] 1) As indicated in the discussion, Lioy et al. (2011) restored many core features (both behaviorally and anatomically) in a MeCP2-deficient mouse with the selective expression of MeCP2 in only astrocytes. In the study, they showed an increase in the number of vGluT1 terminals and restored morphology. Does the persistent effect shown by Rakela et al. depend on the increase in presynaptic release of vGluT1 synapses? If persistently altered for long periods of time, would it be capable to capture either a morphological or histological change in terminals?

Lioy’s findings of morphological changes and elevated vGluT1 levels were based on long-term restoration of MeCP2 in astrocytes in vivo. It is difficult to imagine that similar morphological changes would occur with our astrocyte stimulation that is on the order of tens of seconds, and our resolution in slices limits any detection of fine scale structural changes. Further, our data showing an increase in neurons in the frequency of both GABA and glutamatergic synaptic currents, after astrocyte stimulation, reflects increased release probability and not postsynaptic alterations. We have emphasized this point in the first paragraph of the Discussion. As presynaptic, the increase in glutamatergic responses would therefore reflect release from vGluT1 presynaptic terminals.

2) On a related note, what characteristics of presynaptic release are altered in the face of astrocytic gating? Does the number of release sites increase? The lack of amplitude changes suggests that quantal size is not changing, but how do these changes impact short-term plasticity measures?

We conclude that the effect is presynaptic and not postsynaptic based on the increase in frequency and absence of change in amplitude for synaptic events, as noted by the reviewer. We stress that in our Discussion. We have no data that address changes in release site number or the mechanisms involved in the short term plasticity as the neuronal circuitry involved is likely complicated.

Reviewer #2:[…] 1) For the Par-1 immuno staining, please present some analyses for the expression within astrocytes. Include some average data. The Par-1 response is an important stimulus that is used in this paper, but the reader needs to be convinced that Par-1 receptors exist in cortical astrocytes.

To address this important concern we added a new Figure 3 showing immunohistochemical images comparing the Par1 labeling of astrocytes and neurons. These images make clear that we can detect labeling in astrocytes but not cortical neurons. Therefore, quantitative comparisons were not possible. Par-1 expression in astrocytes is well documented and we included references to this effect. In addition, in Results we have added links to transcriptome analyses done by others showing expression of mRNA in astrocytes. This potential concern, along with reviewer 3’s concern of whether Par-1 expression is strictly astrocytic, is now addressed further in a new part of the Discussion with appropriate references.

2) Please provide a clear reference to a published paper showing that gramicidin pores are not permeable to chloride. Again, changing the Cl^-^ reversal potential is a nice approach used in this study, but the reader needs to be convinced that gramicidin perforated patches behave in the way the authors state. This knowledge is critical to interpret the result in Figure 4 that GABA is excitatory at this stage of development in the cortex. If other people have addressed this topic before then additional references should be provided.

We now cite in the Results section the earliest and best studies showing cation selectivity and the lack of permeation by Cl with gramicidin. The references to an excitatory action played by GABA during early development are in the last paragraph of the Discussion.

3) The presentation of the results is a bit long-winded and ponderous. Perhaps sharpen the text to make it briefer and more to the point.

The overall length changes achieved by consolidation of figures, along with the movement of ponderous pharmacology of basal synaptic currents to supplemental, have resulted in a ‘more to the point’ Results section.

4) The authors make the point clearly that they do not know the identity of the substance released from astrocytes to trigger the altered neuronal signaling. If they are sure it is not glutamate release, then they should state this explicitly. Also, could the substance be a change in ATP, K^+^ and or Ca^+^ in the extracellular fluid? If so, please discuss.

We embraced this criticism and have added more logic behind the time-locked averaging and the use of inwardly and outwardly directed currents to test glutamate and GABA as substances released. We now make it clear in the Results and Discussion that GABA is ruled out on the basis of bidirectional experiments in Figure 6 and that glutamate action through ionotropic receptors is also ruled out in Figure 7. We note in the Discussion that there are other candidates, such as those noted by the reviewers, and add appropriate references, but we do not dwell on the subject because we have nothing to add to an already confusing topic.

5) It is understood that tools to manipulate astrocytes are limited and the authors have had to use strong depolarization, Par-1 agonists and Ca^2+^ uncaging. As experimental tools these are fine, but the authors should guide the readership by stating in the Discussion that it is likely that all three of these methods are non-physiological and that the conclusions should be interpreted with these thoughts in mind. Thus it is hard to imagine a scenario when astrocytes depolarize to +60mV.

As outlined in our response to reviewer 1, we have responded by adding to the Discussion a section related to the use of depolarization and the resulting lack of mechanistic insights. We also make clear that depolarization and calcium uncaging are non-physiological stimuli, but we do take issue with the Par1 activation being non-physiological. It has a known biological ligand, mimicked by TFLLR, and a well characterized signaling pathway. The claim that Par1 is physiological is now better supported by references in the Results and Discussion.

6) Add statistics, p values, n numbers and statistical tests to the key figures so the reader can see what is happening just by looking at the figures.

As requested, statistics, statistical tests used, number of recordings, number of animals and gender have been added to every appropriate figure.

7) Many of the scatter graphs show mean +/- sd. This is fine, but I wonder if it would be better to show box and whisker plots with mean, sd and sem? I suspect some readers will appreciate such graphs more than mean +/- sd.

In some experiments shown the numbers of recordings are not appropriate for SEM and we want to remain consistent throughout the paper with our representation of the results. For this reason, we hope the reviewer will be satisfied with our showing only the mean +/- SD.

Reviewer #3:[…] Figure 2 and histograms elsewhere. These were apparently built from the data from >1 cell, and according to the Results (subsection “Astrocytes mediate increases in cortical neuron signaling”, second paragraph), event frequency varied greatly among cells. How were histograms built to avoid overburdening the distribution with events from high-frequency cells and to thus keep the amplitude distribution representative?

The reviewer raises an important point that we addressed as follows. For every recording, we generated a plot of event number vs. amplitude, similar to the cumulative plot shown in Figure 2. We then normalized each plot, based on peak event number, and summed them. Because each recording is now weighted equally it can be compared statistically to determine whether the pre and post amplitudes are similar, as maintained on the basis of the cumulative plot in Figure 2. We have done this for both Figure 2 and Figure 4 data and find that the pre and post amplitudes are not statistically different. We added both weighted histograms, along with the statistics, as a new supplemental figure.

Figure 2. What is the interpretation of the increase in select large events? The statistical treatment of this is unclear.

The statement was deleted.

Regardless of the pulse amplitude used for depolarizing stimulation, the physiological relevance of the strong depolarization of astrocytes is highly questionable. It is good that the investigators use a second method of local TFLLR application, but the relevance and specificity of this are also questionable (see below). Were other physiologically-based methods of astrocyte stimulation attempted? Were they unsuccessful?

TFLLR was the only physiologically relevant stimulation we tried. We chose this agonist because the receptor is highly enriched in astrocytes and the signaling pathway in astrocytes has been characterized. As outlined in our response to the editor and reviewer 2, we have added references to the considerable Discussion supporting a role for TFLLR in a naturally occurring signaling pathway. The specificity of the Par1 receptor for astrocytes is further elaborated in the discussion now on the basis of immunohistochemistry (replicated in our own hands) and specific inhibition of the neuronal response by dialysis of the astrocyte with BAPTA (the TFLLR response is known to engage a calcium response).

Par1 activation with TFLLR occurs in isolated neurons and causes them to fire action potentials (PMID 21827709). Thus, the possibility of a direct TFLLR on neurons needs to be treated with respect.

Our strongest evidence that TFLLR does not work via neurons is provided by our finding that inclusion of BAPTA in the astrocyte electrode abolishes all activation. Additionally, we detected Par1 by immunohistochemistry in cortical astrocytes but not neurons. We have added a section in the Discussion emphasizing this important issue. We also added to the Discussion a critical evaluation of the paper referenced by the reviewer. That study was restricted to a single class of neurons, shown by the authors to express Par1 on the bases of immunohistochemistry and electrophysiology. However, within that population of neurons, <5% of the cells depolarized in response to TFLLR, and then only after greatly prolonged bath application of >120 sec. The absence of immunohistochemical detection in cortical neurons, the BAPTA block of activation in astrocytes and the differences in time scale for activation between astrocytes and neurons all indicate that Par1 is working through astrocytes in the cortex.

The Ca^2+^ signal initiated by astrocyte depolarization should be documented in the authors' hands, using the brief-pulse protocols used to elicit the PSC frequency increase. A few papers are cited, but voltage-gated Ca^2+^ influx in astrocytes is not universally observed.

We tested for calcium currents using whole cell astrocyte patch clamp recordings. However, the low input resistance and space clamp issue greatly compromised detection. We were also unsuccessful in measuring any effects of depolarization on calcium signals in the soma with calcium-indicating dyes. As noted by this reviewer, the presence of voltage-gated calcium channels remains under dispute, but we believe positive findings from published studies outweigh the negative. To this point, we cite published data showing both T and L type calcium channels in acutely dissociated astrocytes.

The authors seem to want to interpret experiments in Figure 4–Figure 8 as a test of the astrocyte derived signal (e.g. Discussion, second paragraph). However, the experiments testing the nature of gliotransmitter (glutamate, GABA) are confounded by the same substances serving as neurotransmitters. A likely model to explain the data is that substance X, released from astrocytes, increases spiking selectively in principal neurons surrounding the recorded cell (perhaps because only principal cells express the receptor for X). This accounts for the increase in sEPSCs. Local synaptic interactions between principal neurons and interneurons then explain the increase in sIPSCs. When viewed through the lens of this explanation, the experiments designed to determine the polarity of GABA signaling seem tangential.

We did not make sufficiently clear our rationale or logic underlying the pharmacological block experiment. The logic was that if blocking one of the transmitters did *not* block TFLLR activation, then we could rule it out as a potential activator of the neuronal responses, independent of its role in neuronal circuitry. In this way, our results ruled out GABA but not glutamate. The logic and interpretation is now clarified in the Results and Discussion. To further test glutamate, we performed time locked averaging to see if we could detect early synchronized synaptic responses that might reflect direct actions by glutamate (Figure 7). This experiment could only be done with depolarization. We detected responses during the first 200 ms that were initially hopeful, but because they were all eliminated in the presence of TTX we conclude that all of these responses reflect downstream circuitry as proposed by the reviewer. All of these interpretations have been added to Results and Discussion.

The attempts to measure EPSCs and IPSCs by clamping midway between reversal potentials appear fraught. Membrane noise is high in Figure 5, likely because of the depolarized Vm value. This will increase false positives and perhaps false negatives. The false positive problem is highlighted by incomplete antagonism in Figure 5. Overall, it is difficult to believe the quantifications from this protocol.

We agree that the S/N is worse than other measurements where the driving force was larger. However, as explained above, we used this approach principally to test whether GABA or glutamate could represent the activator substance released by the astrocyte. This approach required different directionality of currents. Thus, this approach was not designed to provide exacting estimates of glutamate and GABA contributions beyond that shown by pharmacology of inward currents.

Figure 8. Sample traces should be shown here. I have trouble reconciling the time course data with Figure 2. What am I missing? The statistical treatment here (2 peaks/) is not clear. Further, the basis for the time course data informing the likelihood of glutamate as mediator (subsection “Interdependence of GABA and glutamatergic signaling pathways”, last paragraph) are lost on me.

We understand the confusion. We were unable to show the time course in the same format as previous figures because the pre and post responses were not consecutive for this experiment and the bins were accumulated from 100 trials. However, as a better means of showing time course, we now put pre and post on the same time scales and we have added the requested raw data to clarify (Figure 7). The TTX recording is now clearly indicated as such. We have also added more detail about the protocol and the logic in the Results section.

Figure 12: Raw images that serve as the basis for the Ca^2+^ imaging should be shown. How confident can the authors be that images derive from astrocytes? Wouldn't signals from astrocyte processes be most relevant to the studies? How were signals from neurons/neuron dendrites excluded?

We now include raw images for the Ca imaging in Figure 10. All of the measurements shown were taken from somas of SR101 positive cells to insure they were astrocytes. We agree that measurements from astrocyte processes would be relevant and we spent a great deal of time trying to analyze distal sparks. However, they were not synchronized, making them difficult to quantitate, and further, speaking to the reviewer’s concern, the sparks could not be definitively assigned to astrocytic processes, so we were not able to include the distal signaling in the analyses.

Supplemental Figure 5. Ca^2+^ signals from neurons and astrocytes are shown, but the authors state that Fluo-AM preferentially labels astrocytes.

Actually, we state that Fluo4ff is specific to astrocytes. Fluo4, used in the original figure, labels both astrocytes and neurons in our hands. However, the Fluo4 data has been eliminated so now there should be no confusion.

The results seem to suggest that both Ca increase/handling and downstream signal transmission are altered in MeCP2 astrocytes. This is an interesting conclusion but leaves one hanging, without more mechanistic explanation.

Unfortunately, we are hanging – we have no further mechanistic insights other than that the calcium increase is required but not sufficient. Speculation could represent a very long laundry list of potential candidates.

Other points:Sex of WT mice in initial experiments should be specified.

The sexes of all mice are now indicated for all experiments in the figure legends.

A general description of statistical treatment is given in the Materials and methods. For increased transparency, I would like to see the statistical tests specified in the figure legends for each experiment.

Now, all of the statistical tests and outcomes accompany each figure legend.

Abstract, and perhaps elsewhere: 'astrocyte activation' has a specific connotation that should be avoided in this paper.

We changed all references indicating “astrocyte activation” to “astrocyte stimulation”.

Subsection “Astrocytes mediate increases in cortical neuron signaling”, second paragraph. Why mention the decay information as data not shown? If this is important to the arguments presented, show the data. If it is not important, omit the reference.

The reference is omitted.

Subsection “Astrocytes mediate increases in cortical neuron signaling”, second paragraph. The Results describe astrocyte stimulation as a pulse to +60 mV; the figure legend cites +140 mV.

The total step size was +140 mV as indicated (from -80 mV to +60 mV).

[Editors' note: further revisions were requested prior to acceptance, as described below.]

Reviewer #3:The authors have achieved a more effective and streamlined presentation in this revision. As noted by previous reviews, the main limitation remains the limited mechanistic insight. In addition, some important elements remain unclear.The rationale for the GABA experiments in Figure 5 and Figure 6 seems to be that because GABAA receptor activation is excitatory at this age (Figure 5), GABA/GABAARs are candidates to mediate depolarization of the network and thus the increased sPSC frequency. The rationale and where these experiments are headed should be more clearly explained near the beginning of the description of Figure 5.

The rationale for the GABA experiments is indeed based on Figure 5 showing that GABA is excitatory at this age. This result then becomes the premise for separation of currents on the basis of ion selectivity, which is the best way to track differential effects on the GABA and glutamatergic currents. To that point, Figure 6 shows that the inward and outward currents behave largely as expected on the basis of pharmacology, setting the stage for Figure 7 experiments that would not be possible with currents that are all inwardly-directed. We see no way that we can start discussing the rationale for Figure 6 and Figure 7 at the point where we first show that GABA is excitatory in Figure 5. There we address the rationale called for by this reviewer, as directly, succinctly and clearly as we can to do it, in the first sentence after Figure 5 results: “The differences in ion selectivity provided a direct way to determine the individual contributions of GABA- and glutamate-mediated currents to the recorded synaptic currents”.

I cannot overcome my problems with the design of Figure 6 and thus the interpretations of experiments meant to exclude GABA/glutamate as participants in the signaling cascade. The authors seem to concede that S/N is a problem. This does not lend confidence to the interpretation of either positive or negative results in the figure. As it stands, the lack of full pharmacological block in Figure 6—figure supplement 1 suggests that false positives are a problem. Even more concerning is the high frequency of inward sPSCs in Figure 6 in the presence of NBQX/APV. Raw traces post antagonist may help build confidence. It seems to me that the proper design would have been to use CsCl pipet solution, -70 mV, and pharmacologically isolate GABAA sPSCs and in turn AMPAR sPSCs. This would not have affected GABA's influence on the network.

We appreciate that separating the inward and outward currents the way we did it is not standard, but it is the only way to perform the experiments in Figure 7. Therefore, we deemed it logical and necessary to show the pharmacology using this same approach rather than applying pharmacology on all inward currents. In fact, we had performed the pharmacology both ways, but felt the approaches were redundant, so we only presented the pharmacology on the separated currents for the reason stated above. However, we have in hand all of the standard pharmacology characterization proposed by the reviewer and we have added it as Figure 6—figure supplement 2, along with brief presentation of the statistics and data in the Results section. The results from the two approaches are consistent and both show the persistence of a class of inward currents in the presence of GluR blockers. We hope adding this data will convince the reviewer that the data in Figure 6 does not reflect a poor S/N.

I am still confused about the time course data shown in Figure 7. Here the effect of depolarization fades by 150 ms, while in Figure 1 and Figure 4 the effect remains for 20-30 s.

This reviewer missed the distinction of the paradigm in Figure 7 from that in the other figures, and thus its importance, so we need to clarify. As stated above, Figure 6 definitively rules out GABA as being directly released, since the TFLLR response is not affected in the presence of the GABA antagonist. The same strategy, using a glutamate antagonist, implicates glutamate as a candidate. Therefore, we designed a protocol that would test specifically whether release of any inwardly directed current, including glutamate, was synchronized to the depolarization (i.e. immediate release). The time base in Figure 7 reflects this specific paradigm that is completely different from that in Figure 1 and Figure 4, and all the other figures, that use trains of depolarization. In Figure 7, we use a single depolarization and analyze only the first 200 msec where a directly released transmitter action would be expected. This single action potential is not capable of activating fully the neuronal network in the way that a train of depolarization did. However, addition of TTX showed that the currents shown in Figure 7 did reflect a mild activation of the neuronal network, rather than a directly released glutamate or other ionotropic receptor transmitter. We have modified the Results section once again in an effort in order to provide a clearer rationale for this approach and highlight how this paradigm is different from that in the other figures.